# Introducing FORECAST: The Future Outcome Reasoning and Confidence Assessment Benchmark

**Zhangdie Yuan**
University of Cambridge
zy317@cam.ac.uk

**Zifeng Ding**
University of Cambridge
zd320@cam.ac.uk

**Andreas Vlachos**
University of Cambridge
av308@cam.ac.uk

## Abstract

Forecasting is an important task in many domains. However, existing forecasting benchmarks lack comprehensive confidence assessment, focusing on limited question types, and often consist of artificial questions that do not reflect real-world needs. To address these gaps, we introduce FORECAST (Future Outcome Reasoning and Confidence Assessment), a benchmark that evaluates models' ability to make predictions and their confidence in them. FORECAST spans diverse forecasting scenarios involving Boolean questions, timeframe prediction, and quantity estimation, enabling a comprehensive evaluation of both prediction accuracy and confidence calibration for real-world applications.[1]

## 1 Introduction

Recent advances in large language models (LLMs) have significantly improved their performance across a wide range of natural language processing (NLP) tasks. In parallel to these developments, various benchmarks and datasets have been introduced to effectively assess the capabilities of LLMs, particularly in terms of knowledge and reasoning [Zellers et al., 2019, Guo et al., 2023]. Fact-oriented benchmarks, such as TruthfulQA [Lin et al., 2022], evaluate LLMs on factual correctness, focusing on tasks like retrieving and verifying facts that are already known.

Forecasting is a crucial yet challenging task across various domains, including technology, economics, and public policy. Unlike tasks that rely on retrieving and verifying existing knowledge, forecasting requires predicting plausible outcomes for future events, often under uncertainty and incomplete information. Several datasets have been introduced to evaluate LLMs' forecasting capabilities. ForecastQA [Jin et al., 2020] uses a multiple-choice format where models predict future outcomes, but it lacks confidence assessment. AutoCast [Zou et al., 2022] incorporates confidence intervals but not for its forecasting questions. Other datasets, such as ExpTime [Yuan et al., 2024a], are artificially generated from temporal knowledge graph, focusing on explainability.

All of these aforementioned benchmarks overlook a crucial aspect of forecasting: confidence evaluation. Confidence plays a central role in forecasting, as predictions about unresolved events inherently lack definitive answers at the time they are asked. Predictions made with absolute certainty are undesirable, even if they ultimately prove to be correct, because they fail to account for the uncertain nature of future events. Moreover, miscalibrated confidence can lead to poor decision-making: overconfident yet incorrect forecasts may result in costly errors, while underconfident but accurate predictions can erode trust in the model. Therefore, well-calibrated confidence scores are as crucial as the accuracy of the predictions themselves.

To address these issues we present FORECAST: **F**uture **O**utcome **R**easoning and **C**onfidence **A**ssessment. Gold confidence scores are derived from aggregated human forecasts, reflecting community consensus over time. FORECAST focuses on three distinct types of forecasting questions,

---

[1]FORECAST is publicly available at https://huggingface.co/datasets/MoyYuan/FOReCAst.

as illustrated in Table 1: (1) *Boolean Questions*, such as "Will there be a Frontier AI lab in China before 2026?"; (2) *Timeframe Prediction*, such as "When will OpenAI announce GPT-5?"; and (3) *Quantity Estimation*, such as "How many spacecrafts will land on the moon in 2025?" We conduct experiments using a range of models differing in size, training objectives, and training cutoff times, and explore multiple methods for estimating model confidence. Our results reveal that forecasting remains highly challenging for contemporary LLMs. Their confidence calibration is poor overll, and it has inconsistent correlation with predictive accuracy. While larger models sometimes improve performance, the effect is inconsistent and highly task-dependent.

Table 1: Forecasting questions with their resolutions and confidence scores.

| Type | Question | Resolution | Confidence |
|---|---|---|---|
| Boolean Question | Will there be a Frontier AI lab in China before 2026? | Yes | 0.73 |
| Timeframe Prediction | When will OpenAI announce GPT-5? | 2024-08-01 | 0.85 |
| Quantity Estimation | How many spacecrafts will land on the moon in 2025? | 3 | 0.65 |

## 2 FORECAST: Problem Formulation

In FORECAST a system response consists of (1) a prediction that answers a question given the available information and (2) a confidence score in the prediction. This ensures a comprehensive assessment of forecasting, accounting for both correctness and confidence calibration.

Questions belong to three types. (1) *Boolean Questions*, which ask yes/no questions about the occurrence of future events (sometimes within a certain timeframe). Even if such questions are simple to evaluate, they can still be surprisingly challenging [Clark et al., 2019]. (2) *Timeframe Prediction*, which requires predicting a specific timeframe for an event, and are essential for applications where knowing whether an event will happen or not without a timeframe is insufficient. (3) *Quantity Estimation*, which involves providing numerical estimates related to future events.

Formally, let $Q$ represent a question about a future event, and let $M$ denote a system with access to information up to time $t_{train}$ (e.g., the system's knowledge cutoff point). The objective is for $M$ to produce an answer $A$ in $\mathcal{A}$ and an associated confidence score $C$, where:

$$M(Q) \rightarrow (A, C) \quad where \quad A = \arg\max_{a \in \mathcal{A}} P(X = a | Q, \mathcal{K}(t_{train})) \quad and \quad C \in [0, 1]. \quad (1)$$

Here, $\mathcal{K}(t_{train})$ represents the knowledge accessible to the model up to time $t_{train}$. The answer space $\mathcal{A}$ depends on the type of forecasting question: for *Boolean Questions*, $\mathcal{A} = \{\text{Yes}, \text{No}\}$; for *Timeframe Prediction*, $\mathcal{A}$ consists of a single date in the YYYY-MM-DD format; and for *Quantity Estimation*, $\mathcal{A} = \mathbb{R}$, representing real numbers.

## 3 Evaluating Predictions and Confidence

**Boolean Questions** For Boolean questions, where the answer space is $\mathcal{A} = \{\text{Yes}, \text{No}\}$, prediction performance is evaluated using standard classification metrics, including accuracy and F1-score. Confidence calibration is assessed using a modified version of the Brier score [Brier, 1950], which measures the mean squared error between predicted confidence and the gold confidence provided in the dataset, which we assume is provided, and represents the likelihood of an event occurring. The modified Brier score is defined as:

$$\text{Brier} = \frac{1}{N} \sum_{i=1}^{N} (C_i^{\text{pred}} - C_i^{\text{gold}})^2, \quad (2)$$

where $C_i^{\text{pred}}$ is the model's predicted confidence and $C_i^{\text{gold}}$ is the gold confidence. This modification ensures that models are evaluated based on their ability to match the likelihood of an event, with lower Brier scores indicating better calibration.

**Timeframe Prediction** For timeframe prediction, where the answer space consists of specific dates in the YYYY-MM-DD format, predictive accuracy is measured using absolute day error (ADE). Given a

predicted date $D_i^{\text{pred}}$ and the gold date $D_i^{\text{gold}}$, we compute the normalized error as:

$$E_i^{\text{ADE}} = \frac{2}{1 + e^{-\alpha|D_i^{\text{pred}} - D_i^{\text{gold}}|}} - 1, \tag{3}$$

where $\alpha$ is a scaling factor that controls how sharply large errors are penalized. This transformation ensures that extreme deviations do not disproportionately dominate the evaluation.

For confidence calibration, we rely on the Continuous Ranked Probability Score (CRPS) [Matheson and Winkler, 1976], which is a generalisation of the Mean Absolute Error to probabilistic forecasts, and extend it to compare the predicted probability distribution with a gold distribution. Specifically, we assume that both the predicted and the gold confidence predictions follow Gaussian distributions $\mathcal{N}(D_i^{\text{pred}}, \sigma_i^{\text{pred}})$ and $\mathcal{N}(D_i^{\text{gold}}, \sigma_i^{\text{gold}})$ respectively, where the standard deviations are computed as:

$$\sigma_i^{\text{pred}} = \sigma_{\max} \cdot (1 - C_i^{\text{pred}}) + \sigma_{\min} \cdot C_i^{\text{pred}}, \quad \sigma_i^{\text{gold}} = \sigma_{\max} \cdot (1 - C_i^{\text{gold}}) + \sigma_{\min} \cdot C_i^{\text{gold}}. \tag{4}$$

Here, $C_i^{\text{pred}}$ is the model's predicted confidence for the $i$th question, and $C_i^{\text{gold}}$ is the corresponding gold confidence provided in our dataset. The parameters $\sigma_{\max}$ and $\sigma_{\min}$ define the upper and lower bounds for the standard deviation. Intuitively, when confidence is low ($C \approx 0$), uncertainty is high, leading to $\sigma \approx \sigma_{\max}$, while when confidence is high ($C \approx 1$), uncertainty is low, resulting in $\sigma \approx \sigma_{\min}$. We then compute the CRPS as the integrated squared difference between the cumulative distribution functions (CDFs) of the predicted and gold distributions:

$$\text{CRPS} = \frac{1}{N} \sum_{i=1}^{N} \int \left( F_i^{\text{pred}}(d) - F_i^{\text{gold}}(d) \right)^2 \, dd, \tag{5}$$

where $F_i^{\text{pred}}$ and $F_i^{\text{gold}}$ denote the CDFs of the predicted and gold Gaussian distributions, respectively. A lower CRPS indicates better calibration, as it reflects a closer match between the predicted uncertainty and the uncertainty as specified by the gold confidence.

**Quantity Estimation**  For quantity estimation, where the answer space consists of non-negative real numbers ($\mathcal{A} = \mathbb{R}_{\geq 0}$), we evaluate prediction performance using two error metrics: absolute percentage error (APE) and mean absolute error (MAE). Given a predicted quantity $Q_i^{\text{pred}}$ and the gold quantity $Q_i^{\text{gold}}$, the normalized errors are computed as:

$$E_i^{\text{APE}} = \frac{2}{1 + e^{-\alpha \frac{|Q_i^{\text{pred}} - Q_i^{\text{gold}}|}{Q_i^{\text{gold}} + \epsilon}}} - 1, \quad E_i^{\text{MAE}} = \frac{2}{1 + e^{-\alpha|Q_i^{\text{pred}} - Q_i^{\text{gold}}|}} - 1. \tag{6}$$

Here, $\epsilon$ is a small constant to prevent division by zero, and $\alpha$ is a scaling factor that controls how sharply large errors are penalized, similar to the timeframe prediction evaluation. Confidence calibration is assessed using CRPS, following the same Gaussian assumption as in timeframe prediction. The predicted quantity is modeled as a Gaussian distribution $\mathcal{N}(Q_i^{\text{pred}}, \sigma_i^{\text{pred}})$, and the gold quantity as $\mathcal{N}(Q_i^{\text{gold}}, \sigma_i^{\text{gold}})$. The standard deviations $\sigma_i^{\text{pred}}$ and $\sigma_i^{\text{gold}}$ are computed using the same formulation as in timeframe prediction.

# 4  FORECAST Construction

## 4.1  Data Source and Question Selection

FORECAST is constructed from Metaculus (www.metaculus.com, see examples in Appendix A), an online forecasting platform where users submit forecasts to questions across domains such as politics, economics, health, and technology. Metaculus aggregates these into a community prediction that is continuously updated until shortly before resolution. Each question has predefined resolution criteria, and the platform enforces guidelines to ensure that all predictions comply with them and are human-generated. We discard questions that Metaculus marks as ambiguous or those resolved in ways that make forecasting infeasible—for example, when the outcome falls outside the prediction range or when resolution criteria change after submission. Also, we only include questions with at least 100 forecasts to maintain statistical reliability.

## 4.2 Extracting Confidence from Crowdsourced Forecasts

In an ideal scenario, we would have both the resolutions and associated confidence scores available at the time a forecast question is posed. However, in practice, we only observe questions with their eventual resolutions and crowd-sourced human forecasts. To derive confidence scores we aggregate these human forecasts, which, despite occasional disagreements especially on non-trivial questions, provide informative signals about confidence. Human predictions can be sometimes incorrect, but they can still serve as a valuable proxy for uncertainty, as they reflect the best available reasoning given the information at the time. Metaculus ensures that these forecasts are manually submitted and fall within calibrated ranges, which encourages us to use them as proxies for evaluating model confidence. This is inline with other approaches to benchmark construction on tasks where humans disagree on labels, e.g. ChaosNLI [Nie et al., 2020, Plank, 2022].

Gold confidence in FORECAST is derived from the final Metaculus community prediction just before question resolution. Following Metaculus's established approach for evaluating probabilistic forecasts, [2] we compute a log score relative to a uniform baseline. This transformation ensures that confidence reflects not only correctness, but also the amount of informational gain over random guessing. It provides a consistent interpretation of confidence across tasks and scales, particularly when outcomes lie in continuous spaces. For Boolean questions, where the human-forecasted probability for the correct outcome is $P^{\text{gold}}$, gold confidence is computed as:

$$C^{\text{gold}} = \sigma \left( \frac{\ln P^{\text{gold}} - \ln 0.5}{\ln 2} \right), \tag{7}$$

where $\sigma(x)$ is the sigmoid function. This yields a score of 0.5 for a random-guess forecast ($P^{\text{gold}} = 0.5$), and smoothly increases or decreases with higher or lower confidence.

For timeframe prediction and quantity estimation, where the human-forecasted probability density function (PDF) assigns likelihood to a continuous outcome $x^{\text{gold}}$, gold confidence is computed as:

$$C^{\text{gold}} = \sigma \left( \frac{\ln f(x^{\text{gold}}) - \ln f_{\text{uniform}}}{2} \right), \tag{8}$$

where $f(x^{\text{gold}})$ is the forecasted probability density at the resolved outcome, and $f_{\text{uniform}}$ is the uniform baseline density over the valid outcome range. The denominator of 2 stabilizes the scale and avoids over-amplifying minor density differences. This approach accounts for the resolution granularity of each question and ensures confidence is not inflated in uncertain or diffuse distributions.

## 4.3 Dataset Statistics and Comparison

FORECAST consists of 2256 forecasting questions, spanning domains such as politics, economics, science, and technology. Each question includes a resolved outcome, a gold confidence score, and a final Metaculus community forecast before resolution. The questions cover a broad temporal range, with resolutions spanning from March 2016 to the end of January 2025. To facilitate model development and evaluation, we split the dataset into 65% training, 10% validation, and 25% test. The full dataset statistics is shown in Appendix B. Note that we do not use the training split for model fine-tuning or prompt optimization; all models are evaluated in a one-shot setting.

Table 2 provides a comparison between FORECAST and existing forecasting benchmarks. Compared to prior datasets, FORECAST uniquely emphasizes both forecasting accuracy and confidence calibration, includes a diverse set of forecasting tasks, and is constructed from a well-established crowdsourced platform with rigorous resolution criteria.

**Extensibility and Long-Term Sustainability** FORECAST is built with extensibility and sustainability in mind. Our data collection pipeline is fully automated, enabling continuous updates as new forecasting questions and outcomes become available. Although this release draws from Metaculus due to its high-quality resolution criteria and historical depth, the same pipeline can be adapted to other platforms such as Polymarket, Manifold Markets, or Good Judgment Open. By supporting multiple data sources and maintaining alignment with evolving real-world events, FORECAST provides a sustainable foundation for benchmarking forecasting models over time. This extensibility

---

[2] https://www.metaculus.com/help/scores-faq/

Table 2: Comparison of key features across our benchmark variants, highlighting our evaluation of confidence across different question types.

| Benchmark | Question Types | Natural Questions | Confidence |
|---|---|---|---|
| ForecastQA [Jin et al., 2020] | MCQ | ✓ | ✗ |
| AutoCast [Zou et al., 2022] | Various | ✓ | ✗ |
| ExpTime [Yuan et al., 2024a] | Boolean | ✗ | ✗ |
| FORECAST | Various | ✓ | ✓ |

not only ensures long-term utility, but also enables future work to study model generalization across domains, platforms, and temporal shifts.

# 5 Experiments on FORECAST

## 5.1 Models

We evaluate a diverse set of large language models (LLMs) with varying training data cutoffs, model sizes, and instruction tuning. To analyze the impact of knowledge recency, we group models by family and use the stated month in Table 3 as the assumed cutoff.[3]

Table 3: Models used in our experiments, grouped by family and ordered by training data cutoff date.

| Model Family | Model Variants | Cutoff Date |
|---|---|---|
| GPT-2 [Radford et al., 2019] | GPT-2, GPT-2 XL | 2017-12 |
| Pythia [Biderman et al., 2023] | 14M, 160M, 2.8B | 2020-03 |
| BLOOM [Scao et al., 2023] | 560M, 7B1 | 2021-12 |
| LLaMA [Touvron et al., 2023] | LLaMA-7B | 2022-08 |
| OLMo [Groeneveld et al., 2024] | 1B, 7B, 7B-Instruct | 2023-03 |
| OLMo-2 [OLMo et al., 2024] | 7B, 7B-Instruct | 2023-12 |
| Qwen 2.5 [Yang et al., 2024] | 1.5B-Instruct, 7B-Instruct | 2023-12 |
| LLaMa 3.1 [Grattafiori et al., 2024] | 8B | 2023-12 |

## 5.2 Inference

Each model is evaluated independently using 1-shot in-context learning. For every question, we sample $n = 10$ outputs from the model using temperature-based decoding. Each output includes a prediction $v_i$ and an associated confidence score $c_i$, derived from the normalized logits. Specifically, we compute the log-probability of each generated token using $\log p_{i,j} = \log \mathrm{softmax}(z_{i,j})[x_{i,j}]$, where $z_{i,j}$ is the model's pre-softmax logit vector at position $j$ for sample $i$, and $x_{i,j}$ is the generated token index. The confidence score $c_i$ is then defined as the average log-probability across all tokens: $c_i = \frac{1}{T} \sum_{j=1}^{T} \log p_{i,j}$, where $T$ is the length of the generated sequence. We observe low variance across the $n$ samples, with most models producing consistent predictions and confidence values.

By default, we select the prediction with the highest normalized confidence across all samples, i.e., $\arg\max_i c_i$. This strategy is equivalent to greedy decoding and forms our primary evaluation setup. We also conduct ablations with several alternative aggregation methods: (1) majority vote, which selects the most frequent prediction $\hat{v} = \mathrm{mode}(v_1, \dots, v_n)$ and assigns it the maximum confidence among agreeing outputs: $\max c_i \mid v_i = \hat{v}$; (2) weighted average, which selects $\hat{v}$ and computes the average of the raw (unnormalized) confidences: $\frac{1}{|\mathcal{I}|} \sum_{i \in \mathcal{I}} a_i$, where $\mathcal{I} = i \mid v_i = \hat{v}$ and $a_i$ is the raw logit-based confidence; and (3) logit-mean probability, which computes the geometric mean of raw probabilities: $\exp\left(\frac{1}{n} \sum_{i=1}^{n} \log p_i\right)$, reported alongside $\hat{v}$. For Boolean questions, we additionally evaluate a Bayesian aggregation method, which averages normalized confidences separately for "yes" and "no" predictions: $\bar{c}_{\mathrm{yes}} = \frac{1}{|\mathcal{Y}|} \sum i \in \mathcal{Y} c_i$ and $\bar{c}_{\mathrm{no}} = \frac{1}{|\mathcal{N}|} \sum i \in \mathcal{N} c_i$, where $\mathcal{Y} = \{i \mid v_i = \text{"yes"}\}$ and $\mathcal{N} = \{i \mid v_i = \text{"no"}\}$. Corresponding results are reported in subsection 5.4.

---

[3]The exact cutoff date of Qwen 2.5 is unclear, with sources suggesting dates ranging from October 2023 to December 2023; we conservatively use 2023-12 as the assumed cutoff to avoid potential knowledge leakage.

Prompts are minimal and structured to ensure consistency across models. Each input includes a forecasting question and a single demonstration example. Instruction-tuned models receive an additional natural-language instruction to align with their training paradigm. Full prompt templates and decoding hyperparameters are provided in Appendix C and Appendix D.

## 5.3 Results

Our experiments on the FORECAST dataset reveal that forecasting remains a challenging task for current LLMs, particularly in estimating and expressing uncertainty. While models often achieve reasonable accuracy in their point predictions, their confidence calibration, measured by Brier score and CRPS, shows substantial variability. This indicates that confidence estimation should be treated as a distinct evaluation axis, not inferred from accuracy alone. Table 4 reports results on a shared question set to ensure fair comparison across models, while Table 5 evaluates each model on all questions available up to its respective training cutoff as a view of each model's forecasting potential.

Table 4: Combined forecasting performance for cutoff date 2023-12, ensuring same questions for all models. CRPS (T) denotes the Continuous Ranked Probability Score for Timeframe Prediction, while CRPS (Q) denotes the Continuous Ranked Probability Score for Quantity Estimation.

| Model | Acc. (↑) | F1 (↑) | Brier (↓) | ADE (↓) | CRPS (T) (↓) | APE (↓) | MAE (↓) | CRPS (Q) (↓) |
|---|---|---|---|---|---|---|---|---|
| GPT2 | 0.59 | 0.31 | 0.37 | 1.00 | 1.00 | 0.25 | 0.73 | 0.72 |
| GPT2-XL | 0.71 | 0.30 | 0.42 | 1.00 | 1.00 | 0.04 | 0.69 | 0.64 |
| Pythia-14m | 0.62 | 0.14 | 0.65 | 1.00 | 1.00 | 0.25 | 0.85 | 0.83 |
| Pythia-160m | 0.67 | 0.31 | 0.51 | 1.00 | 1.00 | 0.02 | 0.67 | 0.65 |
| Pythia-2.8b | 0.52 | 0.29 | 0.45 | 1.00 | 1.00 | 0.12 | 0.73 | 0.69 |
| Bloom-560m | 0.52 | 0.38 | 0.41 | 1.00 | 1.00 | 0.03 | 0.67 | 0.65 |
| Bloom-7b1 | 0.68 | 0.36 | 0.30 | 1.00 | 1.00 | 0.11 | 0.78 | 0.75 |
| Llama-7b | 0.54 | 0.27 | 0.55 | 0.96 | 0.94 | 0.06 | 0.62 | 0.59 |
| OLMo-1B | 0.23 | 0.21 | 0.75 | 1.00 | 1.00 | 0.22 | 0.76 | 0.74 |
| OLMo-7B | 0.22 | 0.23 | 0.83 | 1.00 | 1.00 | 0.22 | 0.82 | 0.81 |
| OLMo-7B-Inst | 0.68 | 0.16 | 0.36 | 1.00 | 1.00 | 0.27 | 0.81 | 0.78 |
| OLMo-2-7B | 0.51 | 0.24 | 0.44 | 1.00 | 1.00 | 0.02 | 0.65 | 0.62 |
| OLMo-2-7B-Inst | 0.59 | 0.30 | 0.41 | 0.87 | 0.82 | 0.09 | 0.60 | 0.57 |
| Qwen-2.5-1.5B-Inst | 0.67 | 0.20 | 0.32 | 1.00 | 1.00 | 0.14 | 0.74 | 0.72 |
| Qwen-2.5-7B-Inst | 0.68 | 0.08 | 0.25 | 1.00 | 1.00 | 0.07 | 0.55 | 0.53 |
| Llama-3.1-8B | 0.08 | 0.00 | 0.99 | 1.00 | 1.00 | 0.02 | 0.58 | 0.54 |

Table 5: Forecasting performance across all tasks in FORECAST, evaluated using questions available to each model based on its training cutoff date. As a result, the number of evaluated questions ($N$) may vary across models. Boolean question metrics include Accuracy, F1, and Brier score (higher is better for Accuracy/F1, lower is better for Brier). Quantity Estimation and Timeframe Prediction use lower-is-better metrics. APE: Absolute Percentage Error, MAE: Mean Absolute Error, CRPS: Continuous Ranked Probability Score, ADE: Absolute Days Error.

| Model | Boolean Questions | | | | Quantity Estimation | | | | Timeframe Prediction | | |
|---|---|---|---|---|---|---|---|---|---|---|---|
| | N | Acc. (↑) | F1 (↑) | Brier (↓) | N | APE (↓) | MAE (↓) | CRPS (↓) | N | ADE (↓) | CRPS (↓) |
| GPT2 | 401 | 0.58 | 0.37 | 0.42 | 81 | 0.23 | 0.87 | 0.86 | 26 | 0.99 | 0.99 |
| GPT2-XL | 401 | 0.67 | 0.32 | 0.45 | 81 | 0.03 | 0.75 | 0.72 | 26 | 1.00 | 1.00 |
| Pythia-14m | 343 | 0.59 | 0.14 | 0.62 | 77 | 0.17 | 0.88 | 0.85 | 25 | 1.00 | 1.00 |
| Pythia-160m | 343 | 0.61 | 0.26 | 0.53 | 77 | 0.03 | 0.78 | 0.76 | 25 | 1.00 | 1.00 |
| Pythia-2.8b | 343 | 0.55 | 0.39 | 0.44 | 77 | 0.07 | 0.81 | 0.79 | 25 | 0.97 | 0.96 |
| Bloom-560m | 263 | 0.49 | 0.41 | 0.45 | 43 | 0.05 | 0.73 | 0.71 | 12 | 1.00 | 1.00 |
| Bloom-7b1 | 263 | 0.64 | 0.33 | 0.32 | 43 | 0.06 | 0.75 | 0.72 | 12 | 1.00 | 1.00 |
| Llama-7b | 226 | 0.57 | 0.34 | 0.52 | 32 | 0.06 | 0.64 | 0.61 | 10 | 0.98 | 0.98 |
| OLMo-1B | 188 | 0.23 | 0.16 | 0.75 | 23 | 0.21 | 0.79 | 0.76 | 6 | 1.00 | 1.00 |
| OLMo-7B | 188 | 0.21 | 0.17 | 0.82 | 23 | 0.22 | 0.84 | 0.82 | 6 | 1.00 | 1.00 |
| OLMo-7B-Inst | 188 | 0.65 | 0.14 | 0.38 | 23 | 0.24 | 0.80 | 0.77 | 6 | 1.00 | 1.00 |
| OLMo-2-7B | 145 | 0.51 | 0.24 | 0.44 | 20 | 0.02 | 0.65 | 0.62 | 4 | 1.00 | 1.00 |
| OLMo-2-7B-Inst | 145 | 0.59 | 0.30 | 0.41 | 20 | 0.09 | 0.60 | 0.57 | 4 | 0.87 | 0.82 |
| Qwen-1.5B-Inst | 145 | 0.67 | 0.20 | 0.32 | 20 | 0.14 | 0.74 | 0.72 | 4 | 1.00 | 1.00 |
| Qwen-7B-Inst | 145 | 0.68 | 0.08 | 0.25 | 20 | 0.07 | 0.55 | 0.53 | 4 | 1.00 | 1.00 |
| Llama-3.1-8B | 145 | 0.08 | 0.00 | 0.99 | 20 | 0.02 | 0.58 | 0.54 | 4 | 1.00 | 1.00 |

**Boolean Questions** Table 4 and Table 5 show that while some models reach reasonable accuracy on binary classification tasks, their confidence calibration remains inconsistent. For instance,

`GPT2-XL` achieves a relatively strong accuracy of 0.67 but has a higher Brier score of 0.45, whereas `OLMo-7B-Instruct` achieves a similar accuracy (0.65) with a lower Brier score of 0.38, indicating better alignment between predicted probabilities and actual outcomes. This discrepancy suggests that models may differ not only in their ability to identify the correct answer, but also in how they distribute confidence over binary options.

Importantly, model performance, whether in terms of accuracy or confidence calibration, does not appear to correlate strongly with scale or training data recency. For example, `LLaMA-7B`, a more recent model, shows lower accuracy (0.57) and higher Brier score (0.52) than older models like `GPT2-XL`. Moreover, `LLaMA-3.1-8B` fails entirely on this task, with an accuracy of 0.08 and a Brier score of 0.99. Manual inspection reveals that it often outputs malformed answers (e.g., long strings of zeros) rather than valid "Yes"/"No" responses, suggesting a failure to follow the prompt format.

**Timeframe Prediction**    Table 4 and Table 5 demonstrate that timeframe prediction is especially difficult for current models. Across the board, both ADE and CRPS values remain close to 1.0, indicating little improvement over naive or uninformed baselines. Only `OLMo-2-7B-Instruct` achieves a CRPS noticeably below this range (0.82), but even this is far from well-calibrated. The generally poor performance suggests that temporal forecasting presents challenges beyond simple factual retrieval or classification. This difficulty likely reflects the inherent complexity of temporal prediction, which often requires models to reason over event dynamics, causality, and elapsed time, all of which are underrepresented in standard pretraining corpora, as well as the challenge posed by the broader space of possible answers, which extends far beyond binary classifications. In contrast to Boolean tasks, where resolution criteria are clearly defined and outcomes are binary, timeframe questions require mapping language input to a continuous temporal target, which demands a finer-grained understanding of how events evolve over time. That most models fail to depart meaningfully from baseline-level CRPS indicates that LLMs may lack sufficient temporal inductive biases or exposure to structured time-referenced reasoning.

**Quantity Estimation**    As shown in Table 4 and Table 5, quantity estimation which predicts a numerical outcome exhibits a broader range of performance across models. Some models such as `Pythia-160M` and `GPT2-XL` achieve very low absolute percentage error (APE = 0.02 and 0.04 respectively), indicating strong point prediction capability. However, their CRPS values (0.65 and 0.72) suggest relatively poor calibration. In contrast, `OLMo-2-7B-Instruct`, with a higher APE of 0.08, attains the lowest CRPS (0.57), suggesting it better captures uncertainty even if its point estimates are less precise. This divergence illustrates that point prediction accuracy and confidence calibration are not necessarily aligned. Predicting a correct numerical value does not guarantee that a model's probability distribution around its prediction reflects uncertainty appropriately. Since CRPS penalizes both distance from the true value and over/under-confidence, this metric provides a more comprehensive view of performance in numerical forecasting tasks. These results suggest that even relatively strong predictive models may require additional mechanisms such as structured uncertainty modeling to produce well-calibrated forecasts in quantity estimation settings.

## 5.4   Findings

**Impact of Model Size on Forecasting Performance**    Model size alone does not reliably predict forecasting performance across tasks. As shown in Table 5, within the Pythia family, `Pythia-2.8b` occasionally improves over smaller variants in certain metrics. For example, it achieves lower CRPS than `Pythia-14m` on quantity estimation but these gains are neither large nor consistent. In some cases, smaller models like `Pythia-160M` outperform their larger counterparts in point prediction metrics such as APE. A similar trend appears within the OLMo family: while the `OLMo-2-7B` variants perform competitively in calibration metrics (e.g., CRPS), this is not accompanied by uniform improvements in accuracy or absolute error. These observations suggest that increases in parameter count do not automatically translate to improved forecasting performance. While larger models often benefit from greater representational capacity, the forecasting task, particularly when it involves expressing well-calibrated uncertainty, may place demands that are orthogonal to model size. Specifically, forecasting tasks may require targeted reasoning over time, quantities, or epistemic uncertainty, which are not clearly emphasized during standard pretraining or scaled parameter growth. These findings indicate that effective forecasting may depend more on model behavior during decoding or task-specific adaptation, rather than scale alone.

Table 6: Effect of instruction tuning on forecasting performance across boolean, timeframe, and quantity estimation questions. Metrics include accuracy, F1, and Brier score for binary questions; ADE and CRPS (T) for timeframe prediction; and APE, MAE, and CRPS (Q) for quantity estimation.

| Model | Acc. (↑) | F1 (↑) | Brier (↓) | ADE (↓) | CRPS (T) (↓) | APE (↓) | MAE (↓) | CRPS (Q) (↓) |
|---|---|---|---|---|---|---|---|---|
| OLMo-7B | 0.21 | 0.17 | 0.82 | 1.00 | 1.00 | 0.22 | 0.84 | 0.82 |
| OLMo-7B-Inst | 0.65 | 0.14 | 0.38 | 1.00 | 1.00 | 0.24 | 0.80 | 0.77 |
| OLMo-2-7B | 0.51 | 0.24 | 0.44 | 1.00 | 1.00 | 0.02 | 0.65 | 0.62 |
| OLMo-2-7B-Inst | 0.59 | 0.30 | 0.41 | 0.87 | 0.82 | 0.09 | 0.60 | 0.57 |

**Impact of Instruction Tuning on Forecasting Performance**    Table 6 compares base and instruct-tuned variants of OLMo-7B and OLMo-2-7B across Boolean, timeframe, and quantity forecasting tasks. For Boolean questions, `OLMo-7B-Instruct` achieves higher accuracy (0.65 vs. 0.21) and a lower Brier score (0.38 vs. 0.82), indicating better confidence calibration. In timeframe prediction, the instruct-tuned `OLMo-2-7B-Instruct` improves uncertainty estimation, with an ADE of 0.87 and CRPS of 0.82, compared to 0.9981 and 0.9970 (both rounded to 1.00) for the base `OLMo-2-7B`. For quantity estimation, instruct-tuned models have slightly higher APE but lower MAE and CRPS, suggesting better uncertainty calibration. While these results point to potential benefits of instruction tuning for probabilistic reasoning, the sample size remains limited due to the recency of instruction-tuned models, and the observed gains may not be statistically robust.

Table 7: Different aggregation methods for extracting predictions and confidence from Llama-7B.

| Aggregation Method | N | Acc. (↑) | F1 (↑) | Brier (↓) |
|---|---|---|---|---|
| Majority Vote | 226 | 0.56 | 0.24 | 0.53 |
| Highest Confidence | 226 | 0.57 | 0.34 | 0.52 |
| Weighted Average | 226 | 0.56 | 0.24 | 0.29 |
| Logit Mean Probability | 226 | 0.56 | 0.24 | 0.62 |
| Bayesian Aggregation | 226 | 0.58 | 0.35 | 0.55 |

**Impact of Aggregation Methods on Forecasting Performance**    Table 7 compares different aggregation methods for deriving the final prediction and confidence estimate from the top 10 outputs of `Llama-7B`. Bayesian Aggregation achieves the highest accuracy (0.58) and F1 score (0.35), suggesting it is the most effective at identifying correct outcomes. However, Weighted Average yields a significantly lower Brier score (0.29), indicating superior confidence calibration compared to others. Majority Vote, Highest Confidence, and Logit Mean Probability produce comparable accuracy and F1 scores but have noticeably higher Brier scores, suggesting weaker uncertainty estimation. These results highlight that even when point prediction performance is similar, aggregation methods substantially impact confidence reliability. The challenge remains in developing techniques that optimize both accuracy and calibration simultaneously, emphasizing the importance of uncertainty-aware forecasting. In preliminary experiments, we also explored verbalized confidence [Xiong et al., 2023], prompting the model to explicitly generate a probability (e.g., "I'm 80% confident..."). However, most models either failed to produce enough meaningful numeric estimates or generated inconsistent responses, particularly on harder questions. This highlights a limitation in current instruction-following capabilities when it comes to quantifying uncertainty directly. Therefore, future work includes fine-tuning for calibrated confidence and integrating task-specific priors. Joint modeling of accuracy and uncertainty remains a key challenge.

**Lack of Correlation Between Forecast Date and Model Performance**    One might expect language models to perform better on forecasting questions that are closer to their training cutoff date, since more relevant information may have been available during pretraining. For instance, a model like `GPT2`, with a cutoff in December 2017, would intuitively have an advantage when forecasting 2018 events compared to events in 2020. However, as shown in Appendix E, we observe no clear improvement for near-cutoff questions. In many cases, model performance remains flat or even inconsistent across time, with no systematic degradation as the temporal distance increases.

This observation suggests that forecasting is meaningfully different from retrieval-based tasks such as fact-checking or question answering [Guo et al., 2022]. While those tasks often benefit from training data that includes the target facts or related content, forecasting involves reasoning under

uncertainty about events that had not yet occurred during training. Models are not explicitly trained to perform such reasoning, and pretraining objectives generally do not reward predictions about future developments. The absence of a correlation between cutoff proximity and performance highlights that forecasting requires capabilities beyond knowledge access, including abstraction, probabilistic reasoning, and temporal inference. These factors make the task more complex and reinforce the need for dedicated methods that go beyond static knowledge retrieval. These findings would not be observable under prior benchmarks that lack confidence metrics or diverse forecasting tasks.

## 6 Related Work

Recent forecasting benchmarks focus on event prediction but largely overlook confidence calibration. OpenForecast [Wang et al., 2025] introduces a large-scale dataset for open-ended, multi-step event forecasting but does not assess model confidence or calibration. ForecastBench [Karger et al., 2024] evaluates binary (Yes/No) forecasting by prompting models for direct probability estimates, but since it queries each option independently, the assigned probabilities do not necessarily sum to 1, leading to potential inconsistencies. Both benchmarks advance the study of future event prediction but stop short of systematically evaluating uncertainty quantification, a crucial aspect for reliable deployment in decision-making contexts. Our work complements these efforts by providing a structured setting for evaluating both forecasting accuracy and probabilistic calibration.

Beyond forecasting, several benchmarks assess language models' reasoning and inference capabilities. COPA [Roemmele et al., 2011] evaluates causal reasoning by presenting a premise and two alternatives, requiring models to select the more plausible cause or effect. HellaSwag [Zellers et al., 2019] challenges models with sentence completion tasks that demand commonsense reasoning, where models must choose the most sensible continuation of a given scenario. PRobELM [Yuan et al., 2024b] assesses models' capacity to rank scenarios by plausibility, bridging the gap between factual accuracy and world knowledge. While these benchmarks probe useful reasoning skills, they are not designed for evaluating predictions about unresolved or future outcomes, nor do they assess confidence calibration in a dynamic context.

## 7 Limitations

FORECAST is built entirely from Metaculus, a high-quality forecasting platform with clear resolution criteria and an active user base. This provides a strong foundation for defining and benchmarking probabilistic forecasting with LLMs. However, it does not reflect the full diversity of forecasting formats, domains, or styles seen in platforms like Good Judgment Open or domain-specific settings. Expanding to other platforms is an important next step to test generalization across forecasting cultures. Our scoring formulation aligns confidence with meaningful deviations from uncertainty and is tailored to Metaculus conventions. However, it can be adapted for use with other platforms or scoring protocols. The goal is to maintain an interpretable and consistent signal across all tasks.

Gold confidence labels are derived from community-aggregated forecasts, offering a practical proxy for crowd belief and uncertainty. While this enables scalable evaluation, it may obscure forecaster disagreement and introduce biases from group dynamics or correlated errors. Future work could incorporate richer uncertainty signals, such as forecast distributions, forecaster rationales, or disagreement metrics. Finally, FORECAST focuses on English-language questions. While this is consistent with Metaculus and existing NLP resources, probabilistic reasoning and uncertainty expression can differ across languages and cultures. Extending the benchmark to multilingual settings and modeling temporal adaptation and real-time reasoning are key directions for improving robustness and real-world applicability.

## 8 Conclusion

We introduce FORECAST, a benchmark for evaluating both forecasting accuracy and confidence calibration in language models. Unlike existing datasets, FORECAST explicitly assesses confidence alongside predictions. Our results show that current models struggle with both prediction and well-calibrated confidence, underscoring the need for improved uncertainty estimation and confidence calibration. Our dataset is now publicly available.

## Acknowledgements

All authors of this paper are supported by the ERC grant AVeriTeC (GA 865958). Andreas Vlachos receives further support from the DARPA program SciFy.

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

# A  Example Metaculus Questions

To illustrate how human forecasts evolve over time, we present two questions from different domains:
Q1 is in the business and geopolitics domain and Q2 is in the technology domain.

> *Q1: Will TikTok become available in the US on both the App Store and Google Play before April 5, 2025?*
> *Q2: When will a SpaceX Starship reach orbit?*

For Q1, Figure 1 shows how community forecasts changed over time, while Figure 2 presents the histogram of the final forecast distribution.

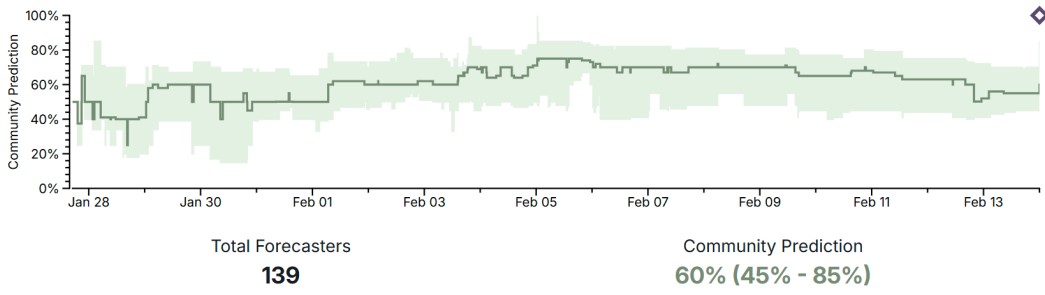

Figure 1: Community prediction trend for a Metaculus question on TikTok's availability in the US.

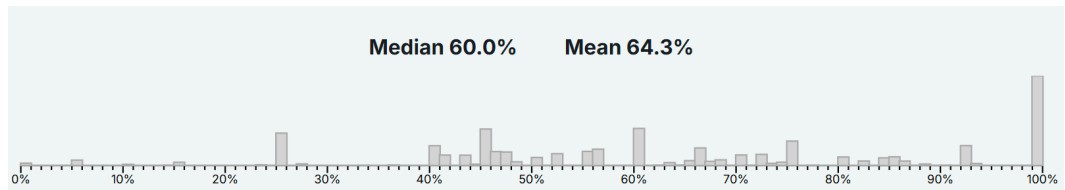

Figure 2: Histogram of final community forecasts.

For Q2, Figure 3 tracks forecast updates, while Figure 4 shows the final probability density function (PDF) of predicted launch dates.

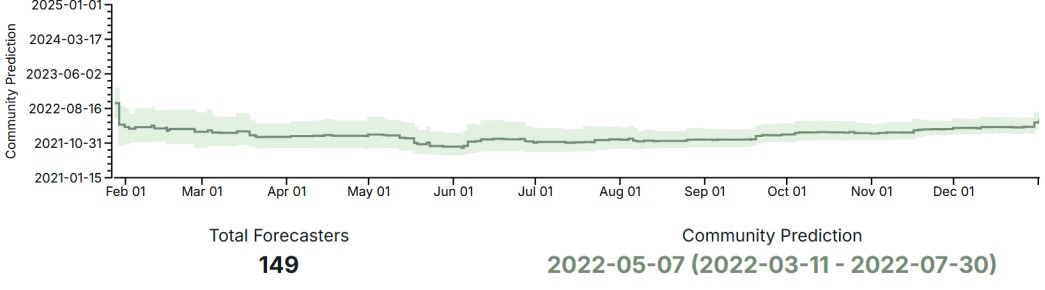

Figure 3: Community prediction trend for SpaceX Starship's first orbital launch.

# B  Dataset Statistics

Table 8 presents detailed dataset statistics, including the total number of questions and their distribution across Boolean Questions, Timeframe Prediction, and Quantity Estimation tasks.

Table 9 provides the top 10 categories and closing months for each task type, illustrating the breadth of topics and temporal diversity in FORECAST.

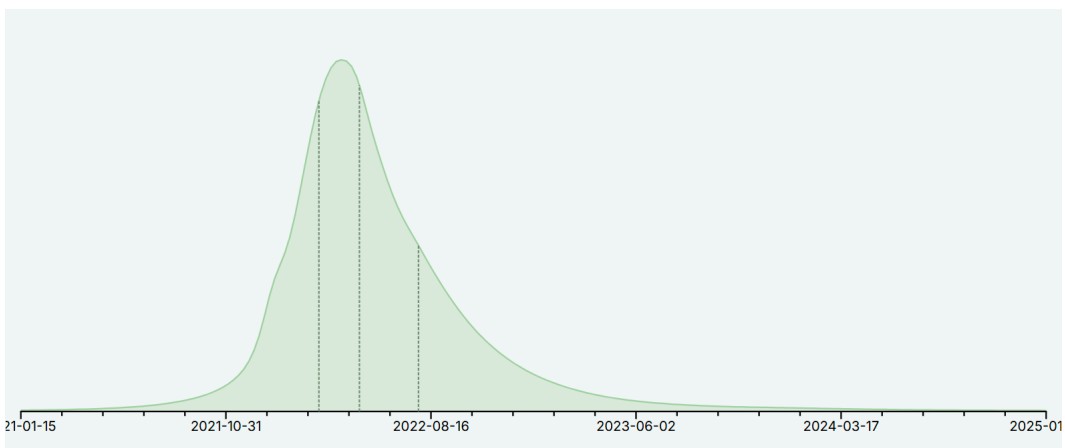

Figure 4: Probability density function of final community forecasts for SpaceX Starship reaching orbit.

Table 8: Dataset statistics for FORECAST, showing the distribution of questions across different forecasting types, with the overall total in the last column.

| Split | Boolean | Timeframe | Quantity | Total |
|---|---|---|---|---|
| Training | 1142 | 90 | 223 | 1465 |
| Validation | 175 | 13 | 35 | 223 |
| Test | 441 | 36 | 91 | 568 |
| **Total** | 1758 | 139 | 349 | 2256 |

## C Prompts

To ensure a fair and consistent evaluation across models, we use simple one-shot prompts with structured outputs in JSON format. For instruction-tuned models, we provide an additional instruction line specifying the task. The prompts are designed for three forecasting question types: Boolean Questions (Yes/No), Timeframe Prediction (YYYY-MM-DD), and Quantity Estimation (numeric values).

Table 9: Top 10 months and categories across the full dataset and each question type. Note that questions can belong to multiple categories, reflecting the diversity of domains represented.

| Global Distribution | | | Timeframe Prediction | | |
|---|---|---|---|---|---|
| **Month** | Count | % | **Month** | Count | % |
| 2023-12 | 187 | 8.29% | 2021-05 | 9 | 6.47% |
| 2024-12 | 105 | 4.65% | 2021-03 | 7 | 5.04% |
| 2022-12 | 89 | 3.95% | 2022-12 | 6 | 4.32% |
| 2021-01 | 65 | 2.88% | 2024-01 | 6 | 4.32% |
| 2024-01 | 62 | 2.75% | 2022-02 | 6 | 4.32% |
| 2024-05 | 56 | 2.48% | 2021-04 | 5 | 3.60% |
| 2025-01 | 49 | 2.17% | 2024-05 | 4 | 2.88% |
| 2023-01 | 48 | 2.13% | 2022-09 | 4 | 2.88% |
| 2020-11 | 47 | 2.08% | 2021-12 | 4 | 2.88% |
| 2020-03 | 41 | 1.82% | 2021-01 | 4 | 2.88% |
| **Category** | Count | % | **Category** | Count | % |
| Politics | 571 | 17.03% | Health & Pandemics | 40 | 18.10% |
| Geopolitics | 486 | 14.50% | Economy & Business | 31 | 14.03% |
| Economy & Business | 476 | 14.20% | Politics | 26 | 11.76% |
| Health & Pandemics | 313 | 9.34% | Technology | 23 | 10.41% |
| Technology | 253 | 7.55% | Computing and Math | 21 | 9.50% |
| Elections | 218 | 6.50% | Geopolitics | 19 | 8.60% |
| Computing and Math | 192 | 5.73% | Artificial Intelligence | 17 | 7.69% |
| Artificial Intelligence | 151 | 4.50% | Space | 13 | 5.88% |
| Natural Sciences | 139 | 4.15% | Law | 7 | 3.17% |
| Law | 104 | 3.10% | Natural Sciences | 6 | 2.71% |
| Quantity Estimation | | | Boolean Questions | | |
| **Month** | Count | % | **Month** | Count | % |
| 2021-02 | 25 | 6.96% | 2023-12 | 167 | 9.50% |
| 2022-12 | 21 | 5.85% | 2024-12 | 94 | 5.35% |
| 2020-04 | 19 | 5.29% | 2022-12 | 62 | 3.53% |
| 2023-12 | 18 | 5.01% | 2024-01 | 51 | 2.90% |
| 2024-05 | 17 | 4.74% | 2021-01 | 45 | 2.56% |
| 2020-03 | 17 | 4.74% | 2025-01 | 44 | 2.50% |
| 2021-01 | 16 | 4.46% | 2023-01 | 38 | 2.16% |
| 2020-11 | 16 | 4.46% | 2024-05 | 35 | 1.99% |
| 2021-10 | 12 | 3.34% | 2023-06 | 34 | 1.93% |
| 2020-12 | 12 | 3.34% | 2024-11 | 29 | 1.65% |
| **Category** | Count | % | **Category** | Count | % |
| Economy & Business | 102 | 22.57% | Politics | 488 | 18.22% |
| Health & Pandemics | 88 | 19.47% | Geopolitics | 428 | 15.98% |
| Politics | 57 | 12.61% | Economy & Business | 343 | 12.80% |
| Geopolitics | 39 | 8.63% | Technology | 205 | 7.65% |
| Artificial Intelligence | 27 | 5.97% | Elections | 190 | 7.09% |
| Elections | 25 | 5.53% | Health & Pandemics | 185 | 6.91% |
| Technology | 25 | 5.53% | Computing and Math | 151 | 5.64% |
| Computing and Math | 20 | 4.42% | Natural Sciences | 124 | 4.63% |
| Environment & Climate | 16 | 3.54% | Artificial Intelligence | 107 | 3.99% |
| Sports & Entertainment | 13 | 2.88% | Law | 96 | 3.58% |

### C.1 Instruction-Tuned Models

For models containing "Instruct" in their name, we use the following prompts:

**Quantity Estimation**

```
You are an AI assistant providing precise numerical forecasts.
Answer the following question with a single numeric value in JSON format.

Example:
Q: How much global photovoltaic energy generation was deployed
   by the end of 2020?
A: { "value": 738 }

Q: $question
A: { "value": "
```

**Timeframe Prediction**

```
You are an AI assistant providing precise date forecasts.
Answer the following question with a single date in YYYY-MM-DD format in JSON.

Example:
Q: When did an AI system achieve a significant victory against
   a professional human in Starcraft 2?
A: { "value": "2019-01-24" }

Q: $question
A: { "value": "
```

**Boolean Questions**

```
You are an AI assistant providing binary (Yes/No) answers.
Answer the following question with "Yes" or "No" in JSON format.

Example:
Q: Will we confirm evidence for megastructures orbiting the
   star KIC 8462852?
A: { "value": "No" }

Q: $question
A: { "value": "
```

### C.2 Base Models

For non-instruction-tuned models, we use the same examples but without additional instructions:

**Quantity Estimation**

```
Q: How much global photovoltaic energy generation was deployed
   by the end of 2020?
A: { "value": 738 }

Q: $question
A: { "value": "
```

**Timeframe Prediction**

```
Q: When did an AI system achieve a significant victory against
   a professional human in Starcraft 2?
A: { "value": "2019-01-24" }
```

```
Q: $question
A: { "value": "
```

**Boolean Questions**

```
Q: Will we confirm evidence for megastructures orbiting the
   star KIC 8462852?
A: { "value": "No" }

Q: $question
A: { "value": "
```

# D   Hyperparameter Settings

## D.1   Generation Hyperparameters

We generate responses using temperature-based sampling with the following hyperparameters:

- `max_length` = 200
- `do_sample` = True
- `top_k` = 50
- `top_p` = 0.9

Among the generated outputs, we select the one with the highest confidence as the final prediction. All experiments are conducted using full precision on an NVIDIA RTX 8000 GPU.

## D.2   Evaluation Hyperparameters

The scaling factor $\alpha$ in Equation 3 and Equation 6 is set to 0.05. For Equation 4, we set $\sigma_{\max}$ to 30 and $\sigma_{\min}$ to 1 for Timeframe Prediction, while for Quantity Estimation, $\sigma_{\max}$ is 20 and $\sigma_{\min}$ is 1. These values ensure that evaluation metrics appropriately scale errors and confidence calibration.

# E   Additional Results

This section provides extended results categorized by the training data cutoff date of each model. Forecasting performance depends on model architecture, scale, and knowledge recency, so we evaluate models with different cutoff dates to examine how access to more recent information influences prediction accuracy and confidence calibration.

Models trained after certain event resolutions may have indirectly encountered outcome-related information, potentially affecting evaluation fairness. This should be considered when interpreting results.

Detailed model-specific performance metrics for Boolean Questions, Timeframe Prediction, and Quantity Estimation are presented in Table 10 to Table 14.

These results highlight trends in forecasting accuracy and confidence calibration across models with different knowledge recency.

# F   Robustness, Ablations, and Methodology Details

This appendix provides further details on our benchmark methodology, robustness checks, and expanded ablations to supplement the main paper.

Table 10: Combined forecasting performance for cutoff date 2017-12. CRPS (T) denotes the Continuous Ranked Probability Score for Timeframe Prediction, while CRPS (Q) denotes the Continuous Ranked Probability Score for Quantity Estimation.

| Model | Acc. (↑) | F1 (↑) | Brier (↓) | ADE (↓) | CRPS (T) (↓) | APE (↓) | MAE (↓) | CRPS (Q) (↓) |
|---|---|---|---|---|---|---|---|---|
| GPT2 | 0.58 | 0.37 | 0.42 | 0.99 | 0.99 | 0.23 | 0.87 | 0.86 |
| GPT2-XL | 0.67 | 0.32 | 0.45 | 1.00 | 1.00 | 0.03 | 0.75 | 0.72 |
| Pythia-14m | 0.60 | 0.15 | 0.62 | 1.00 | 1.00 | 0.18 | 0.86 | 0.84 |
| Pythia-160m | 0.61 | 0.25 | 0.53 | 1.00 | 1.00 | 0.03 | 0.76 | 0.74 |
| Pythia-2.8b | 0.53 | 0.37 | 0.46 | 0.97 | 0.96 | 0.08 | 0.79 | 0.77 |
| Bloom-560m | 0.48 | 0.42 | 0.48 | 1.00 | 1.00 | 0.06 | 0.77 | 0.75 |
| Bloom-7b1 | 0.63 | 0.32 | 0.35 | 0.97 | 0.95 | 0.05 | 0.76 | 0.74 |
| Llama-7b | 0.55 | 0.37 | 0.53 | 0.94 | 0.93 | 0.08 | 0.72 | 0.70 |
| OLMo-1B | 0.23 | 0.20 | 0.79 | 1.00 | 1.00 | 0.15 | 0.84 | 0.82 |
| OLMo-7B | 0.21 | 0.17 | 0.80 | 1.00 | 1.00 | 0.27 | 0.88 | 0.87 |
| OLMo-7B-Inst | 0.67 | 0.33 | 0.39 | 0.92 | 0.91 | 0.29 | 0.80 | 0.78 |
| OLMo-2-7B | 0.54 | 0.33 | 0.50 | 0.92 | 0.91 | 0.04 | 0.70 | 0.68 |
| OLMo-2-7B-Inst | 0.59 | 0.42 | 0.40 | 0.84 | 0.83 | 0.13 | 0.69 | 0.67 |
| Qwen-1.5B-Inst | 0.67 | 0.32 | 0.32 | 1.00 | 1.00 | 0.07 | 0.73 | 0.71 |
| Qwen-7B-Inst | 0.68 | 0.26 | 0.26 | 1.00 | 1.00 | 0.08 | 0.63 | 0.61 |
| Llama-3.1-8B | 0.05 | 0.02 | 0.96 | 1.00 | 1.00 | 0.03 | 0.70 | 0.68 |

Table 11: Combined forecasting performance for cutoff date 2020-03. CRPS (T) denotes the Continuous Ranked Probability Score for Timeframe Prediction, while CRPS (Q) denotes the Continuous Ranked Probability Score for Quantity Estimation.

| Model | Acc. (↑) | F1 (↑) | Brier (↓) | ADE (↓) | CRPS (T) (↓) | APE (↓) | MAE (↓) | CRPS (Q) (↓) |
|---|---|---|---|---|---|---|---|---|
| GPT2 | 0.59 | 0.38 | 0.41 | 0.99 | 0.99 | 0.23 | 0.89 | 0.87 |
| GPT2-XL | 0.67 | 0.32 | 0.45 | 1.00 | 1.00 | 0.03 | 0.77 | 0.74 |
| Pythia-14m | 0.59 | 0.14 | 0.62 | 1.00 | 1.00 | 0.17 | 0.88 | 0.85 |
| Pythia-160m | 0.61 | 0.26 | 0.53 | 1.00 | 1.00 | 0.03 | 0.78 | 0.76 |
| Pythia-2.8b | 0.55 | 0.39 | 0.44 | 0.97 | 0.96 | 0.07 | 0.81 | 0.79 |
| Bloom-560m | 0.50 | 0.44 | 0.46 | 1.00 | 1.00 | 0.05 | 0.78 | 0.77 |
| Bloom-7b1 | 0.63 | 0.33 | 0.35 | 0.97 | 0.95 | 0.05 | 0.79 | 0.76 |
| Llama-7b | 0.55 | 0.37 | 0.53 | 0.94 | 0.93 | 0.06 | 0.72 | 0.70 |
| OLMo-1B | 0.23 | 0.19 | 0.78 | 1.00 | 1.00 | 0.16 | 0.87 | 0.85 |
| OLMo-7B | 0.20 | 0.15 | 0.80 | 1.00 | 1.00 | 0.26 | 0.89 | 0.87 |
| OLMo-7B-Inst | 0.66 | 0.30 | 0.40 | 0.92 | 0.91 | 0.27 | 0.80 | 0.78 |
| OLMo-2-7B | 0.53 | 0.33 | 0.51 | 0.92 | 0.90 | 0.03 | 0.71 | 0.68 |
| OLMo-2-7B-Inst | 0.59 | 0.41 | 0.40 | 0.83 | 0.82 | 0.12 | 0.70 | 0.68 |
| Qwen-1.5B-Inst | 0.66 | 0.32 | 0.33 | 1.00 | 1.00 | 0.07 | 0.75 | 0.73 |
| Qwen-7B-Inst | 0.68 | 0.27 | 0.26 | 1.00 | 1.00 | 0.07 | 0.64 | 0.62 |
| Llama-3.1-8B | 0.05 | 0.02 | 0.96 | 1.00 | 1.00 | 0.02 | 0.71 | 0.69 |

### F.1 Robustness of Gold Confidence

A crucial consideration for our benchmark is the robustness of the gold confidence labels, which are derived from aggregated Metaculus community forecasts. We implemented several empirical checks and quality controls to mitigate potential issues such as "group-think" or correlated errors.

- Empirical Agreement: To empirically evaluate the reliability of the community predictions, we computed Cohen's $\kappa$ between the Metaculus community's binary predictions (defined as "Yes" probability > 50%) and the actual resolved outcomes on our Boolean questions. This yielded $\kappa = 0.676$, which indicates substantial agreement according to widely-used interpretation scales. This result suggests the aggregated forecasts strongly reflect true underlying uncertainties.

- Platform Calibration: Metaculus itself routinely evaluates and publicly reports its calibration quality.[4] These analyses demonstrate that community predictions closely align with observed outcome frequencies, with approximately half of the observed outcome rates lying within the 90% credible interval around the ideal calibration line. This strong empirical calibration supports our use of these forecasts as a practical proxy for ground-truth confidence.

---

[4]https://www.metaculus.com/questions/track-record

Table 12: Combined forecasting performance for cutoff date 2021-12. CRPS (T) denotes the Continuous Ranked Probability Score for Timeframe Prediction, while CRPS (Q) denotes the Continuous Ranked Probability Score for Quantity Estimation.

| Model | Acc. (↑) | F1 (↑) | Brier (↓) | ADE (↓) | CRPS (T) (↓) | APE (↓) | MAE (↓) | CRPS (Q) (↓) |
|---|---|---|---|---|---|---|---|---|
| GPT2 | 0.61 | 0.35 | 0.38 | 1.00 | 1.00 | 0.26 | 0.85 | 0.84 |
| GPT2-XL | 0.67 | 0.26 | 0.45 | 1.00 | 1.00 | 0.03 | 0.74 | 0.70 |
| Pythia-14m | 0.60 | 0.13 | 0.65 | 1.00 | 1.00 | 0.22 | 0.86 | 0.84 |
| Pythia-160m | 0.63 | 0.24 | 0.52 | 1.00 | 1.00 | 0.02 | 0.73 | 0.70 |
| Pythia-2.8b | 0.52 | 0.33 | 0.44 | 1.00 | 1.00 | 0.07 | 0.77 | 0.74 |
| Bloom-560m | 0.50 | 0.44 | 0.45 | 1.00 | 1.00 | 0.05 | 0.73 | 0.71 |
| Bloom-7b1 | 0.63 | 0.33 | 0.32 | 1.00 | 1.00 | 0.06 | 0.75 | 0.72 |
| Llama-7b | 0.57 | 0.36 | 0.52 | 0.99 | 0.98 | 0.05 | 0.65 | 0.62 |
| OLMo-1B | 0.24 | 0.18 | 0.77 | 1.00 | 1.00 | 0.22 | 0.84 | 0.82 |
| OLMo-7B | 0.22 | 0.18 | 0.80 | 1.00 | 1.00 | 0.23 | 0.85 | 0.84 |
| OLMo-7B-Inst | 0.67 | 0.18 | 0.40 | 0.92 | 0.92 | 0.22 | 0.76 | 0.74 |
| OLMo-2-7B | 0.53 | 0.27 | 0.49 | 0.93 | 0.92 | 0.03 | 0.67 | 0.64 |
| OLMo-2-7B-Inst | 0.61 | 0.38 | 0.39 | 0.86 | 0.84 | 0.10 | 0.64 | 0.61 |
| Qwen-1.5B-Inst | 0.66 | 0.24 | 0.33 | 1.00 | 1.00 | 0.10 | 0.75 | 0.72 |
| Qwen-7B-Inst | 0.68 | 0.14 | 0.26 | 1.00 | 1.00 | 0.08 | 0.60 | 0.59 |
| Llama-3.1-8B | 0.06 | 0.00 | 0.95 | 1.00 | 1.00 | 0.02 | 0.62 | 0.59 |

Table 13: Combined forecasting performance for cutoff date 2022-08. CRPS (T) denotes the Continuous Ranked Probability Score for Timeframe Prediction, while CRPS (Q) denotes the Continuous Ranked Probability Score for Quantity Estimation.

| Model | Acc. (↑) | F1 (↑) | Brier (↓) | ADE (↓) | CRPS (T) (↓) | APE (↓) | MAE (↓) | CRPS (Q) (↓) |
|---|---|---|---|---|---|---|---|---|
| GPT2 | 0.62 | 0.36 | 0.36 | 1.00 | 1.00 | 0.26 | 0.81 | 0.80 |
| GPT2-XL | 0.68 | 0.27 | 0.44 | 1.00 | 1.00 | 0.03 | 0.74 | 0.70 |
| Pythia-14m | 0.61 | 0.16 | 0.65 | 1.00 | 1.00 | 0.19 | 0.87 | 0.85 |
| Pythia-160m | 0.65 | 0.25 | 0.53 | 1.00 | 1.00 | 0.02 | 0.73 | 0.70 |
| Pythia-2.8b | 0.52 | 0.32 | 0.45 | 1.00 | 1.00 | 0.08 | 0.78 | 0.74 |
| Bloom-560m | 0.50 | 0.40 | 0.43 | 1.00 | 1.00 | 0.05 | 0.72 | 0.71 |
| Bloom-7b1 | 0.65 | 0.33 | 0.31 | 1.00 | 1.00 | 0.08 | 0.78 | 0.74 |
| Llama-7b | 0.57 | 0.34 | 0.52 | 0.98 | 0.98 | 0.06 | 0.64 | 0.61 |
| OLMo-1B | 0.23 | 0.18 | 0.78 | 1.00 | 1.00 | 0.22 | 0.83 | 0.80 |
| OLMo-7B | 0.23 | 0.17 | 0.82 | 1.00 | 1.00 | 0.22 | 0.87 | 0.86 |
| OLMo-7B-Inst | 0.65 | 0.12 | 0.40 | 0.91 | 0.90 | 0.21 | 0.78 | 0.75 |
| OLMo-2-7B | 0.54 | 0.26 | 0.47 | 0.91 | 0.91 | 0.03 | 0.68 | 0.65 |
| OLMo-2-7B-Inst | 0.60 | 0.33 | 0.40 | 0.93 | 0.90 | 0.10 | 0.65 | 0.62 |
| Qwen-1.5B-Inst | 0.68 | 0.25 | 0.31 | 1.00 | 1.00 | 0.11 | 0.76 | 0.74 |
| Qwen-7B-Inst | 0.68 | 0.12 | 0.25 | 1.00 | 1.00 | 0.10 | 0.60 | 0.58 |
| Llama-3.1-8B | 0.06 | 0.00 | 0.96 | 1.00 | 1.00 | 0.02 | 0.62 | 0.58 |

- Quality-Control Filters: We explicitly implemented quality-control measures to ensure aggregation robustness. We excluded all questions with fewer than 100 unique forecasters, thereby greatly reducing the risk that a small, uninformed, or correlated group could disproportionately influence the crowd estimate.

- Aggregation Methodology: The Metaculus platform employs skill-weighted, time-decayed aggregation of individual predictions. This mechanism reduces the influence of less accurate or historically less-calibrated forecasters, further mitigating the impact of unreliable forecasts.

- Forecaster Incentives: While not a traditional prediction market, Metaculus does incorporate monetary stakes through partnerships and paid tournament incentives. These mechanisms provide extrinsic motivation and help align forecasters' incentives with predictive accuracy.

## F.2 LLaMA-3.1-8B Failure Mode Analysis

Our results in Section 5.3 showed LLaMA-3.1-8B performing very poorly on Boolean questions (0.08 accuracy). A closer audit revealed this was a semantic failure, not a syntactic one.

- Schema Compliance: The model adhered to the requested JSON schema in > 99% of cases.

Table 14: Combined forecasting performance for cutoff date 2023-03. CRPS (T) denotes the Continuous Ranked Probability Score for Timeframe Prediction, while CRPS (Q) denotes the Continuous Ranked Probability Score for Quantity Estimation.

| Model | Acc. (↑) | F1 (↑) | Brier (↓) | ADE (↓) | CRPS (T) (↓) | APE (↓) | MAE (↓) | CRPS (Q) (↓) |
|---|---|---|---|---|---|---|---|---|
| GPT2 | 0.60 | 0.36 | 0.38 | 1.00 | 1.00 | 0.26 | 0.74 | 0.72 |
| GPT2-XL | 0.65 | 0.24 | 0.42 | 1.00 | 1.00 | 0.04 | 0.71 | 0.65 |
| Pythia-14m | 0.60 | 0.16 | 0.64 | 1.00 | 1.00 | 0.22 | 0.85 | 0.82 |
| Pythia-160m | 0.63 | 0.26 | 0.50 | 1.00 | 1.00 | 0.02 | 0.69 | 0.66 |
| Pythia-2.8b | 0.52 | 0.34 | 0.44 | 1.00 | 1.00 | 0.11 | 0.76 | 0.72 |
| Bloom-560m | 0.53 | 0.42 | 0.41 | 1.00 | 1.00 | 0.07 | 0.70 | 0.67 |
| Bloom-7b1 | 0.66 | 0.39 | 0.30 | 1.00 | 1.00 | 0.10 | 0.79 | 0.76 |
| Llama-7b | 0.55 | 0.32 | 0.51 | 1.00 | 1.00 | 0.06 | 0.60 | 0.57 |
| OLMo-1B | 0.23 | 0.16 | 0.75 | 1.00 | 1.00 | 0.21 | 0.79 | 0.76 |
| OLMo-7B | 0.21 | 0.17 | 0.82 | 1.00 | 1.00 | 0.22 | 0.84 | 0.82 |
| OLMo-7B-Inst | 0.65 | 0.14 | 0.38 | 1.00 | 1.00 | 0.24 | 0.80 | 0.77 |
| OLMo-2-7B | 0.51 | 0.24 | 0.44 | 1.00 | 1.00 | 0.02 | 0.65 | 0.62 |
| OLMo-2-7B-Inst | 0.59 | 0.33 | 0.41 | 0.91 | 0.87 | 0.12 | 0.61 | 0.57 |
| Qwen-1.5B-Inst | 0.65 | 0.20 | 0.34 | 1.00 | 1.00 | 0.13 | 0.72 | 0.69 |
| Qwen-7B-Inst | 0.68 | 0.12 | 0.25 | 1.00 | 1.00 | 0.10 | 0.56 | 0.54 |
| Llama-3.1-8B | 0.07 | 0.00 | 0.96 | 1.00 | 1.00 | 0.02 | 0.59 | 0.55 |

- Semantic Issue: Instead of outputting "Yes" or "No", the model frequently outputted a floating-point numeral. For example, the string `"0.0"` followed by 381 zero digits appeared in 35.6% of its Boolean responses. This behavior was model-specific; no other models in our evaluation exhibited this phenomenon. This suggests the model checkpoint may prefer to express probability mass directly rather than committing to a discrete label.

- Post-processing Result: To understand the model's underlying capability, we ran a simple post-processing step: we treated these numeric strings as confidence estimates and mapped them to binary labels using a 0.5 threshold (e.g., $< 0.5 \rightarrow$ "No", $\geq 0.5 \rightarrow$ "Yes"). Under this mapping, LLaMA-3.1-8B achieves 0.682 accuracy, a score comparable to other models. This clarification contextualizes the original low score, attributing it to a specific failure in instruction-following rather than a fundamental lack of forecasting ability.

## F.3 Generality of Aggregation Ablation

In Section 5.4, we demonstrated that aggregation methods significantly impact performance for LLaMA-7B. We selected LLaMA-7B as it offered a favorable balance between model recency and a large available test set (N=226) for robust analysis.

To further support the generality of this finding, we ran the same ablation on BLOOM-7B, evaluated on 263 Boolean questions. The results, shown in Table 15, confirm that the aggregation method is a meaningful modeling choice. Brier scores varied by over 0.4, and categorical metrics (Accuracy, F1) also shifted. This reinforces that aggregation is not a neutral implementation detail but a critical factor affecting both accuracy and calibration.

Table 15: Aggregation method ablation for BLOOM-7B on Boolean questions (N=263).

| Model | Aggregation Method | Accuracy | F1 | Brier |
|---|---|---|---|---|
| BLOOM-7B1 | Majority Vote | 0.71 | 0.23 | 0.37 |
| BLOOM-7B1 | Highest Confidence | 0.64 | 0.33 | 0.32 |
| BLOOM-7B1 | Weighted Average | 0.71 | 0.23 | 0.29 |
| BLOOM-7B1 | Logit Mean Probability | 0.71 | 0.23 | 0.71 |
| BLOOM-7B1 | Bayesian Aggregation | 0.64 | 0.33 | 0.49 |

## F.4 Temperature Sensitivity Analysis

A methodological concern is the effect of inference-time sampling temperature on the distribution of predictions.

- Default Temperature Rationale: Throughout our experiments, we deliberately adopted the default sampling temperature specified in each model's configuration (e.g., `generation_config.json`). This is 1.0 for most models (GPT-2, Pythia, BLOOM, LLaMA, OLMo), 0.7 for Qwen-2.5 variants, and 0.6 for LLaMA-3.1-8B. This approach aligns our setup with the developers' intended settings and is standard practice in LLM evaluation for ensuring reproducibility and neutrality.

- Clarification: In an earlier discussion, "temperature scaling" was mistakenly mentioned as an aggregation method in Section 5.4. We clarify that this method was not used; the evaluation of temperature sensitivity was considered a separate robustness check.

- Robustness Analysis: To directly address sensitivity, we conducted an additional analysis on LLaMA-7B, repeating our primary evaluation using five temperatures: 0.1 (low-entropy), 0.7, 0.9, 1.0 (default), and 1.5 (high-entropy).

The results are summarized in Table 16. The metric shifts are generally modest and well within the range of performance differences observed across different models.

Table 16: Temperature sensitivity analysis for LLaMA-7B across all metrics.

| Temp. | Acc. ($\uparrow$) | F1 ($\uparrow$) | Brier ($\downarrow$) | ADE ($\downarrow$) | CRPS (T) ($\downarrow$) | APE ($\downarrow$) | MAE ($\downarrow$) | CRPS (Q) ($\downarrow$) |
|---|---|---|---|---|---|---|---|---|
| 1.0 (default) | 0.54 | 0.27 | 0.55 | 0.96 | 0.94 | 0.06 | 0.62 | 0.59 |
| 0.1 | 0.56 | 0.13 | 0.64 | 1.00 | 1.00 | 0.02 | 0.60 | 0.55 |
| 0.7 | 0.53 | 0.22 | 0.44 | 1.00 | 1.00 | 0.09 | 0.72 | 0.68 |
| 0.9 | 0.54 | 0.23 | 0.53 | 1.00 | 1.00 | 0.04 | 0.66 | 0.61 |
| 1.5 | 0.50 | 0.29 | 0.52 | 1.00 | 1.00 | 0.17 | 0.71 | 0.68 |

This sweep shows that while temperature does perturb metrics (e.g., Brier score shifts $\approx \pm 0.09$), the changes are slight and far less significant than the differences between models. Timeframe metrics remained pinned near 1.0, indicating negligible influence. This confirms that using the default temperature is a representative and methodologically sound choice.

## F.5 Rationale for Gaussian Distribution Assumption

For continuous tasks (Timeframe and Quantity), we model predictive uncertainty using a Gaussian distribution to calculate the Continuous Ranked Probability Score (CRPS). This choice was based on two primary reasons:

1. Empirical Realism: Many continuous-valued targets in FORECAST, such as inflation rates or economic growth, are known to approximately follow a normal distribution, particularly after trends are removed. This assumption has long-standing support in macroeconomic and climatological modeling.

2. Aggregation Theory: Even if individual forecasters' internal distributions are not Gaussian, the Central Limit Theorem implies that aggregating a large number of independent forecasts will naturally yield an approximately normal distribution. Our filter requiring $\geq 100$ unique forecasters helps ensure this assumption is practical, improving the reliability of the aggregated distributions.

## F.6 Rationale for Static Prediction Snapshot

We acknowledge that forecast distributions evolve as new information emerges. Our evaluation, however, deliberately adopts a static snapshot methodology for practical reasons.

We chose forecast snapshots very close to each question's resolution date (typically within 24 hours of resolution). This strategy, illustrated in Figure 1, draws from the far-right end of the forecast time-series, representing a stable, well-informed state of the crowd's belief. These late-stage forecasts exhibit minimal volatility and integrate the most recent, comprehensive information. Many real-world applications rely on such "last-minute" forecasts.

While explicitly modeling the *temporal dynamics* of forecasting (i.e., how models update predictions) is an important and valuable future research direction, it is beyond the scope of this paper's static evaluation goal.

