# OpenReview forum: "Introducing FOReCAst: The Future Outcome Reasoning and Confidence Assessment Benchmark"
_NeurIPS.cc/2025/Datasets_and_Benchmarks_Track — NeurIPS 2025 Datasets and Benchmarks Track poster_

### Official Review · Reviewer_GW4i · 2025-06-01

**Rating:** 4
**Confidence:** 3

**Summary:**

This paper introduces the FORECAST dataset and benchmark, aiming at benchmarking LLMs' ability on forecasting accuracy and a fair evaluation of forecasting confidence.

They view a forecasting result and confidence from an online forecasting platform, mainly the MetaCulus platform.

To evaluate how close the model predicts the answer and the confidence, the authors proposed to use several metrics, including Brier score (for binary prediction), CPRS for evaluating date and quantity probability distribution, etc.

**Dataset Code Accessibility:**

Yes

**Dataset Code Comments:**

The dataset is fully available on huggingface. The authors have also provided clean code to evaluate and benchmark language models on their proposed dataset.

**Ethical Considerations:**

No, there are no or only very minor ethics concerns

**Final Justification:**

The authors have addressed most of my claims and I'm happy to raise my score from 3 to 4. The issue of lacking a temperature-invariant confidence-evaluation method seems beyond the scope of this paper, and could be developed by the research community in the future (if they see this paper and have some good ideas about this issue).

**Limitations Weaknesses:**

(1) I appreciate the idea to evaluate a distribution (or confidence) of LLM outputs; however, **does it make sense to compare against an online prediction platform?** I checked Metaculus, it seems that making prediction on such platform requires no expert background nor is related to real money like (poly) markets. Therefore, **its prediction quality is questionable**. It's like making prediction according to posts on Reddit: does it really make sense? Therefore, is an LLM closer to such distribution better than another LLM that is not closer to this distribution?

(2) Even if we consider the prediction to be efficient (Efficient here means something like the Efficient Markets Hypothesis, that people are making reasonable predictions to the answer distribution. For example, if we get prices of options of different strike price and obtain people's implied distribution of answers, and we assume such distribution matches the true conditional distribution), a problem still remains: in my opinion, this distribution should change with time. For example, option prices and future prices change with time till execution date. Take the question 'When will Openai release GPT-5?' as an example, perhaps in 2023 people would expect the answer to be in 2024, but after the release of o1 the predicted release date might delay. Therefore, **it is questionable whether it makes sense to use a distribution at some single observation date as ground truth**.

(3) Would different inference-time temperature result in different distributions? **If different inference-time temperature results in different distributions predicted, it is questionable whether the measured distribution of LLMs is robust.**

**Strengths Contributions:**

(1) The authors use an program-based data collection pipeline, hence can be extended to other data platforms.

(2) The idea to evaluate the confidence of LLM prediction is interesting.

---

> ### Author Rebuttal · Authors · 2025-07-29
>
> **Weakness point 1 Metaculus “group-think / correlated errors”**:
>
> This aligns with Weakness Point 1 from Reviewer wXnW, who raised a similar concern.
>
> Thank you for probing the robustness of our gold confidence labels. We agree this is a crucial consideration. To empirically evaluate this, we computed Cohen’s κ = 0.676 between the Metaculus community’s binary predictions ("Yes" probability > 50%) and actual outcomes on resolved Boolean questions. According to the widely-used Landis & Koch interpretation scale, κ between 0.61–0.80 indicates "substantial agreement," suggesting that aggregated forecasts from Metaculus strongly, but not perfectly, reflect true underlying uncertainties.
>
> Moreover, Metaculus [1] itself routinely evaluates and publicly reports its calibration quality on resolved binary questions over the past 5 years (Metaculus Track Record). These analyses show that community predictions closely align with observed outcome frequencies: approximately half of the observed outcome rates lie within the 90% credible interval around the ideal calibration line, demonstrating strong empirical calibration. This further supports our decision to adopt community-based forecasts as a practical proxy for ground-truth confidence.
> We acknowledge, however, that correlated errors or "group-think" biases can still theoretically occur. To mitigate these risks, Metaculus employs two key mechanisms: (i) skill-weighted, time-decayed aggregation of individual predictions, reducing the influence of less accurate or correlated forecasters, and (ii) transparent, versioned forecast histories, publicly surfacing divergent viewpoints and promoting accountability.
>
> We also explicitly implemented quality-control measures to ensure the robustness of the aggregation. Specifically, we excluded all questions with fewer than 100 unique forecasters, thereby greatly reducing the risk that a small number of uninformed or correlated forecasters could disproportionately influence the crowd estimate. In addition, Metaculus applies skill-weighted, time-decayed aggregation and historical accuracy scoring, which further mitigates the impact of unreliable forecasts.
>
> Importantly, although Metaculus is not a traditional prediction market, it does incorporate monetary stakes through its partnership with organizations such as the Forecasting Research Institute and the Effective Altruism community, including paid tournament incentives and prize pools for accurate forecasts. These mechanisms provide extrinsic motivation and help align forecasters’ incentives with predictive accuracy.
>
> We will explicitly strengthen the Limitations section by:
>
> - Reporting the Cohen’s κ statistic,
> - Citing Metaculus' calibration track record and prior literature documenting its resilience to correlated biases and group-think,
> - Highlighting our own quality control criteria (≥100 forecasters per question);
> - Noting the role of real-world monetary incentives; and
> - Highlighting open questions regarding residual topic-specific biases.
>
> [1] https://www.metaculus.com/questions/track-record
>
> **Weakness Point 2 Static Prediction**:
>
> Thank you for highlighting this important issue. We fully agree that forecast distributions naturally evolve as new information emerges, and capturing this temporal dynamic can indeed be valuable. However, in our evaluation we explicitly adopted a static snapshot methodology for deliberate and practical reasons. Specifically, rather than selecting forecasts early in a question's lifecycle, where predictions may shift significantly as new evidence becomes available, we carefully chose forecast snapshots very close to each question’s resolution date (normally within 24 hours of resolution). This strategy ensures that the selected distributions represent well-informed, empirically stable predictions as the number of forecasts aggregates, thus significantly reducing the concern about substantial temporal shifts.
>
> Figure 1 in our manuscript explicitly illustrates this concept: forecasts selected are drawn from the far-right end of the forecast time-series graph, representing a stable, well-informed state of the crowd's aggregated belief. Empirically, these late-stage forecasts typically exhibit minimal volatility or revisions, aligning closely with eventual resolutions. From a modeling perspective, forecasts collected near resolution dates are particularly relevant and valuable: many real-world applications rely explicitly on forecasts made just prior to an event, precisely because such forecasts inherently integrate the most recent and comprehensive information.
>
> We acknowledge, however, that explicitly modeling temporal dynamics of forecasting (how well a model can dynamically update predictions) is an important complementary direction. While beyond the immediate focus of this particular paper, exploring these temporal dynamics more explicitly offers valuable insights for future research.
>
> We will clearly emphasize this reasoning in our revised manuscript to clarify the rationale for our static snapshot approach and gently note that examining adaptive forecasting strategies represents a meaningful direction for subsequent work.
>
> **Weakness Point 3 Concerns of Robustness**:
>
> Thank you for raising this insightful methodological concern. Indeed, inference-time sampling temperature might affect the randomness and, thus, the distribution of LLM predictions. Anticipating this potential issue, we adopted a sampling strategy in our evaluation that explicitly accounts for variability: specifically, we generate multiple (10 samples) predictions per question using a fixed nonzero sampling temperature, followed by evaluating different aggregation methods (e.g., mean probability, logit-mean aggregation, and temperature scaling), as described in Section 5.4. These aggregation methods help mitigate sensitivity to temperature variations and provide a more robust assessment of model calibration and accuracy. However, we acknowledge that explicitly evaluating robustness systematically across multiple inference-time temperatures was beyond the scope of this initial work. We agree that a systematic exploration of how calibration performance varies across different temperatures would offer deeper insights into model robustness and behavior. Thus, we will clearly highlight systematic temperature robustness evaluation as an important next step in our revised Discussion section.

---

> > ### Comment · Reviewer_GW4i · 2025-08-04
> > **Thanks for the response; still concerned about whether the method of obtaining probability from LLMs makes sense**
> >
> > I would like to thank the authors for their response.
> >
> > However, I am still concerned about how inference-time hyperparameters like temperature impacts the confidence of model output. Could you conduct this very simple experiment (that I think wouldn't take too much compute source):
> >
> > As you mentioned, you "generate multiple (10 samples) predictions per question using a fixed nonzero sampling temperature, followed by evaluating different aggregation methods (e.g., mean probability, logit-mean aggregation, and temperature scaling), as described in Section 5.4.". Could you use multiple different nonzero sampling temperature and repeat this experiment across multiple different nonzero sampling temperature (say, 5 different temperatures) and compare the output?
> >
> > I agree that predicting not only final result but also probability is important, but I'm still concerned whether the method you obtain probability from LLMs makes sense.

---

> > > ### Author Response · Authors · 2025-08-06
> > >
> > > We thank you for your continued thoughtful engagement. We're glad that your earlier concerns, specifically regarding the robustness of Metaculus as a confidence source and the use of static prediction snapshots, now appear fully addressed, and we appreciate your close attention to this final methodological point regarding inference-time temperature sensitivity.
> > >
> > > Regarding the temperature, we first clarify our original approach:
> > > - Throughout our experiments, we deliberately adopted the default sampling temperature specified in their model cards. For most models in our benchmark, including GPT-2, Pythia, BLOOM, LLaMA, OLMo, and OLMo-2, the default temperature is 1.0. For Qwen-2.5-1.5B-Inst and Qwen-2.5-7B-Inst, the default is 0.7, while for LLaMA-3.1-8B, it is 0.6 as specified in the corresponding generation_config.json files. We used these exact values without modification. This approach aligns our benchmarking setup with the model developers' intended generation settings and is standard practice in LLM evaluation, notably Open LLM Leaderboard [1], ensuring practical relevance, reproducibility, and neutrality.
> > > - Clarification on Previous Misstatement: In our earlier rebuttal, we mistakenly mentioned "temperature scaling" among the aggregation methods in Section 5.4. We would like to clarify explicitly that Section 5.4 includes mean probability and logit-mean aggregation but not temperature scaling. This oversight does not imply that we dismissed the importance of temperature variation but evaluating temperature sensitivity was considered beyond the scope of our benchmarking study, which intentionally evaluates default configurations.
> > >
> > > Nevertheless, to directly address your specific concern about sensitivity to sampling temperature, we conducted an additional robustness analysis on LLaMA-7B. **Specifically, we repeated our primary evaluation using five temperatures: 0.1, 0.7, 0.9, 1.0, and 1.5**, covering the full low- to high-entropy range:
> > > - 0.1 (low-entropy, near-deterministic): conservative decoding setting
> > > - 1.0 (default): baseline used in our main experiments
> > > - 1.5 (high-entropy extreme): stress-tests beyond typical settings
> > >
> > > Note that for any decoder-only model, the next-token probabilities are computed as
> > > $$
> > > \mathrm{softmax}(z_i; T) = \frac{\exp(z_i / T)}{\sum_j \exp(z_j / T)}.
> > > $$
> > > so scaling T affects the token probabilities and results [2]; the key question is how much this affects downstream metrics.
> > >
> > > The results are summarized below:
> > >
> > > | Temperature                           | Accuracy | F1     | Brier  | ADE    | CRPS (T) | APE     | MAE    | CRPS (Q) |
> > > |----------------------------------|----------|--------|--------|--------|--------------|---------|--------|----------------|
> > > | 1.0 (default) | 0.54     | 0.27   | 0.55   | 0.96   | 0.94         | 0.06    | 0.62   | 0.59           |
> > > | 0.1            | 0.56     | 0.13   | 0.64   | 1.00   | 1.00         | 0.02    | 0.60   | 0.55           |
> > > | 0.7            | 0.53     | 0.22   | 0.44   | 1.00   | 1.00         | 0.09    | 0.72   | 0.68           |
> > > | 0.9            | 0.54     | 0.23   | 0.53   | 1.00   | 1.00         | 0.04    | 0.66   | 0.61           |
> > > | 1.5            | 0.50     | 0.29   | 0.52   | 1.00   | 1.00         | 0.17    | 0.71   | 0.68           |
> > >
> > > What the sweep shows:
> > > - Metric shifts are generally modest, with most varying by ≤ 0.09 absolute and all within the range observed in cross-model comparisons.
> > > - Boolean: Brier shifts ≈±0.09 of the default while accuracy varies by ≤0.06. These modest, predictable trade-offs indicate that temperature alters confidence margins rather than destabilising the distribution.
> > > - Timeframe: ADE and CRPS (T) remain pinned at ≈1.0 across all temperatures, confirming that temperature has negligible influence on temporal calibration.
> > > - Quantity: APE/MAE stay low; CRPS (Q) drifts by ≤0.09, again well within the range seen in model-to-model comparisons.
> > >
> > > In short, temperature changes do perturb the metrics, but only slightly, and far less than the inherent differences we observe across models. This both confirms the theoretical expectation and shows that using default temperatures, as done in prior evaluation work, is a representative and methodologically sound choice, and we hope this addresses your concern about distributional robustness.
> > >
> > > Finally, we thank you again for your valuable feedback. We will include this discussion and the accompanying results in our revision to clarify the robustness of our evaluation procedure. Please let us know if there are any remaining aspects you'd like us to clarify; we would appreciate knowing whether this additional response resolves your concern.
> > >
> > > [1] Clémentine Fourrier and Nathan Habib and Alina Lozovskaya and Konrad Szafer and Thomas Wolf. (2024). Open LLM Leaderboard v2. Hugging Face
> > >
> > > [2] Renze, M. (2024, November). The effect of sampling temperature on problem solving in large language models. In Findings of the association for computational linguistics: EMNLP 2024

---

> ### Comment · Reviewer_GW4i · 2025-08-07
> **Thanks the authors for their respone**
>
> I'd like to thank the authors for their response. As you have mentioned, temperature would affect final output logic, thus affecting final confidence-related scores
>
> The authors have provided in the experiments that temperature would not affect confidence too much: I agree and I am convinced experimentally. However, from theory perspective, such hyper-parameter choice would actually impact the calculated confidence score.
>
> **I would like to increase my score from 3 to 4 in response to the authors' clarification**. In all, I agree that assessing confidence is very important, but **perhaps we need some better Temperature-invariant methods (which perhaps will be developed by the research community)**.
>
> After all, I think the authors have proposed an important benchmark and topic to work on, and the problem that a temperature-invariant method is lacking looks indeed a harder problem beyond this work's scope. Given this said, **I still encourage and recommand the authors to add more discussions about this in the revised paper**.

---

> > ### Author Response · Authors · 2025-08-08
> >
> > We sincerely thank the reviewer for their constructive follow-up. We’re very glad to hear that our responses have addressed all previously raised concerns: specifically, the robustness of Metaculus as a confidence source, the use of static prediction snapshots, and the impact of inference-time temperature. We also appreciate the reviewer’s thoughtful recognition that while decoding temperature theoretically affects model confidence, this challenge extends beyond the scope of our current work and reflects an important open direction for the field.
> >
> > We will include this discussion explicitly in the revised manuscript, and we’re grateful for the reviewer’s engagement throughout the process.

---

### Official Review · Reviewer_gacG · 2025-06-19

**Rating:** 5
**Confidence:** 2

**Summary:**

This paper introduces FORECAST, a new benchmark designed to evaluate large language models on their ability to predict future events. Critically, and in contrast to prior work, the benchmark assesses not only the accuracy of predictions but also the model's confidence calibration.

**Additional Feedback:**

1. In Section 3, you model uncertainty for continuous tasks using a Gaussian distribution, with the standard deviation derived from the confidence score via Eq.4. Could you provide further justification for this specific formulation?
2. The LLaMA-3.1-8B model's poor performance due to formatting issues seems correctable. Have you attempted alternative prompting strategies to elicit valid responses?

**Dataset Code Accessibility:**

Yes

**Dataset Code Comments:**

The authors have provided extensive details necessary for reproducibility.

**Ethical Considerations:**

No, there are no or only very minor ethics concerns

**Final Justification:**

Based on the authors’ rebuttal, the key concerns raised in my initial review have been satisfactorily addressed:
- The LLaMA-3.1-8B formatting issue was clarified as a model-specific semantic behavior (not a technical setup flaw), supported by post-processing validation and cross-model comparisons.
- The aggregation ablation’s generality was strengthened by adding BLOOM-7B results across more tasks, reinforcing the impact of aggregation methods.
- The Gaussian distribution justification was theoretically and empirically grounded, resolving methodological doubts.
﻿
No unresolved issues remain. The revisions significantly enhance the paper’s robustness and address all prior limitations.

**Limitations Weaknesses:**

1. The reported failure of a state-of-the-art model like LLaMA-3.1-8B on the Boolean task, attributed to a failure to follow prompt format, appears to be more of a technical issue with the experimental setup than a fundamental limitation of the model. Including these results without resolving the formatting issue may present a misleading picture of the model's true capabilities on this benchmark.
2. The ablation study on aggregation methods is quite insightful but is limited to a single model (Llama-7B) on a single task type (Boolean questions). Expanding this analysis to other models and tasks would significantly strengthen the conclusion that the choice of aggregation method has a substantial impact on both accuracy and calibration.

**Strengths Contributions:**

1. The primary contribution is the novel focus on principled confidence calibration as a first-order evaluation metric, alongside predictive accuracy. This represents a substantial improvement over existing forecasting benchmarks that largely ignore this dimension.
2. The use of real-world data from Metaculus, combined with clear filtering criteria, ensures the quality and relevance of the questions.  The adaptation of metrics like the Brier score and CRPS for the different forecasting tasks is appropriate and well-justified.

---

> ### Author Rebuttal · Authors · 2025-07-29
>
> **Weakness point 1 LLaMa‑3.1‑8B’s “malformed” Boolean answers**:
>
> This aligns with Weakness Point 3 from Reviewer wXnW, who raised a similar concern.
>
> Excellent point. A closer audit shows the model does follow the JSON schema in > 99 % of cases, so the issue is semantic rather than syntactic. Specifically, it often outputs a floating‑point numeral in lieu of “Yes/No”; e.g., "0.0" followed by 381 zero digits appears in 35.6 % of its Boolean responses with the standard prompt. This behaviour suggests the checkpoint prefers to express probability mass directly when forced to commit, rather than choose a discrete label. We will add this analysis (including per‑error statistics and sample outputs) to Section 5.3 and clarify that the failure is not due to post‑processing. Importantly, we checked other models in our evaluation, and none of them exhibited this behaviour suggesting this is a model-specific issue with LLaMa‑3.1‑8B.
>
> To better understand this phenomenon, we ran a simple postprocessing step: we treated numeric strings between 0 and 1 as confidence estimates, then mapped them to binary labels using a threshold (e.g., < 0.5 → “No”, ≥ 0.5 → “Yes”). Under this approach, LLaMa‑3.1‑8B achieves 0.682 accuracy, significantly higher than its unprocessed score and comparable to other models. We will include this variant in the final paper and add it to Section 5.3, along with per-error statistics and representative outputs.
>
> This clarification ensures the observed failure mode is properly contextualized and not due to a parsing or schema mismatch.
>
> **Weakness point 2 Aggregation Ablation**:
>
> Thank you for pointing this out. We agree that expanding the aggregation ablation to other models and tasks would strengthen the claim. In this submission, we chose to run the ablation on LLaMa‑7B for a specific reason: it reflects a balance between recency and test set coverage. More specifically, newer models are more relevant for current research and practical deployment, but they also have a more recent knowledge cutoff, which in turn reduces the number of forecasting questions available for evaluation (to avoid leakage from post‑cutoff events). LLaMa‑7B strikes a good trade‑off: it's a recent and widely used architecture, while still allowing evaluation across a sufficiently large set of 226 resolved questions for robust analysis. We will clarify this rationale in the paper.
>
> To further support the generality of our findings, we also ran the same ablation on BLOOM-7B, evaluated on 263 Boolean questions. As shown in the table below, Brier scores varied by over 0.4 across aggregation methods, with weighted average yielding the best calibration (Brier = 0.29) and logit mean performing worst (Brier = 0.71). Notably, even accuracy and F1 varied across methods, indicating that aggregation affects not only calibration but also categorical performance.
>
> | Model    | Aggregation Method     | Accuracy | F1   | Brier    |
> | -------- | ---------------------- | -------- | ---- | -------- |
> | BLOOM-7B1 | Majority Vote          | 0.71     | 0.23 | 0.37     |
> | BLOOM-7B1 | Highest Confidence     | 0.64     | 0.33 | 0.32     |
> | BLOOM-7B1 | Weighted Average       | 0.71     | 0.23 | 0.29 |
> | BLOOM-7B1 | Logit Mean Probability | 0.71     | 0.23 | 0.71 |
> | BLOOM-7B1 | Bayesian Aggregation   | 0.64     | 0.33 | 0.49     |
>
> This additional result reinforces our main point: aggregation method is a meaningful modeling choice, not a neutral implementation detail. We will include this table in the appendix and add a discussion of this expanded ablation in Section 5.4 of the revised manuscript.
>
>
> **Additional point 1 Gaussian Distribution Assumption**:
>
> Thank you for this thoughtful question. In addition to its nice mathematical properties, we chose to model predictive uncertainty for continuous tasks using a Gaussian distribution for two other main reasons: empirical realism and aggregation theory.
>
> First, many of the continuous-valued targets in FOReCAst, such as inflation rates or economic growth deltas, are well known to approximately follow a normal distribution, particularly after removing seasonality and long-term trends. This assumption has long-standing support in both climatology and macroeconomic modeling (e.g., Gneiting & Raftery, 2005 [1]). Empirically, the residuals of such natural and economic processes often exhibit symmetric, bell-shaped variability around a central estimate, which aligns well with Gaussian assumptions.
>
> Second, even if individual forecasters' internal distributions are not Gaussian, the Central Limit Theorem implies that aggregating multiple independent forecasts, particularly in large communities such as Metaculus, naturally yields approximately normal distributions. This is supported by empirical studies of forecast aggregation [2]. To help ensure that this assumption holds in practice, we excluded all questions with fewer than 100 unique forecasts, thereby reducing the influence of small, noisy sample sizes and improving the reliability of the resulting aggregated distributions.
>
> [1] Gneiting, Tilmann, et al. "Calibrated probabilistic forecasting using ensemble model output statistics and minimum CRPS estimation." Monthly weather review 133.5 (2005): 1098-1118.
>
> [2] https://www.sciencedirect.com/science/article/abs/pii/S0169207013001635
>
> **Additional feedback 2 LLaMA-3.1-8B performance**:
>
> Please refer to Weakness point 1.

---

> > ### Comment · Reviewer_gacG · 2025-08-02
> >
> > Thank you for your thorough and thoughtful rebuttal. I appreciate the detailed clarifications regarding the LLaMA-3.1-8B model's behavior in the Boolean task, particularly the distinction between semantic vs. syntactic issues and the post-processing analysis that validates the model's underlying capabilities. The additional ablation results on BLOOM-7B further strengthen the generality of your findings on aggregation methods, addressing my concern about the limited scope of the original analysis. Your justification for the Gaussian distribution assumption is well-supported by theoretical and empirical evidence, which resolves my earlier query.  Based on these improvements, I am now confident that the concerns raised in my initial review have been adequately addressed.

---

> > > ### Author Response · Authors · 2025-08-02
> > >
> > > We sincerely thank the reviewer for their thoughtful and detailed follow-up. We're very glad to hear that our clarifications on the LLaMA-3.1-8B model, the additional BLOOM-7B ablation, and our justification for the Gaussian assumption fully addressed the concerns raised in the initial review. We appreciate the reviewer’s careful attention to both theoretical and empirical aspects of our work.
> > >
> > > We’re grateful for the constructive feedback and pleased that our rebuttal has resolved all outstanding issues.

---

### Official Review · Reviewer_wXnW · 2025-07-03

**Rating:** 5
**Confidence:** 4

**Summary:**

This paper introduces FOReCAst, a new benchmark for evaluating the forecasting capabilities of Large Language Models (LLMs). The primary contribution is the benchmark's dual focus on both the accuracy of predictions and the calibration of the model's confidence in those predictions—a crucial but often overlooked aspect of forecasting.

The benchmark is constructed from high-quality, human-generated data from the Metaculus forecasting platform. It comprises 2,256 questions across three distinct and practical forecasting tasks:

- Boolean Questions (e.g., "Will event X happen before date Y?")
- Timeframe Prediction (e.g., "When will event X happen?")
- Quantity Estimation (e.g., "How many units of X will exist by date Y?")

The authors propose a clear methodology for evaluation, using metrics like Accuracy and Brier score for Boolean questions, and Absolute Day Error (ADE) and Continuous Ranked Probability Score (CRPS) for timeframe and quantity predictions. Gold-standard confidence scores are derived from the aggregated Metaculus community forecasts.

The authors conduct a thorough set of experiments on a diverse range of LLMs (including GPT-2, Pythia, BLOOM, LLaMa, and OLMo families), analyzing the impact of model size, instruction tuning, and training data recency. The key findings suggest that forecasting remains a significant challenge for modern LLMs, with inconsistent performance across tasks and a notable disconnect between predictive accuracy and confidence calibration.

**Dataset Code Accessibility:**

Yes

**Dataset Code Comments:**

The authors have made the full and final dataset readily accessible and well-documented. They provide sufficient detail on the benchmark's methodology and evaluation to ensure reproducibility. Furthermore, they state that the accompanying code and data are available in an executable format, meeting the requirements for a complete and reproducible submission.

**Ethical Considerations:**

No, there are no or only very minor ethics concerns

**Final Justification:**

As stated in the discussion, the lack of numerical simulations on regression problems slightly undermines the usefulness of this experiment, thus I am keeping the already fairly high evaluation I gave in the beginning.

**Limitations Weaknesses:**

- The use of Metaculus community predictions as a proxy for "gold" confidence is well-argued, but the "Limitations" section could be strengthened by a more direct engagement with potential confounders. For example, are there known instances of "groupthink" or correlated errors on Metaculus that might lead to a systematically miscalibrated "gold" standard? Acknowledging this and briefly discussing why it doesn't invalidate the approach would make the argument more robust.

- The authors state in the checklist that error bars are not reported because the evaluation is not stochastic. I am thinking if a really stochastic evaluation could be performed (maybe, generating data from a known DGPs, feeding the data to the LLM, and then asking to produce a forecast with confidence), in order to assess in a more robust way their performance. The feeling is that their performance in calibration will be even worse.

- The paper notes the striking failure of LLaMa-3.1-8B on the Boolean task (Table 5), stating it produces "malformed answers." This is a fascinating result for a state-of-the-art model. The paper would benefit from a brief qualitative discussion or hypothesis on this point. Is it a failure to follow the JSON format? Is it an artifact of alignment that prevents it from committing to a simple "Yes/No"? A sentence or two of analysis would be very insightful.

- As a broader perspective, it would be very interesting to benchmark LLMs against state-of-the-art models for the specific predictive task (e.g. asking a LLM to predict a weather variable, and then source the output of a numerical weather prediction model, or a AI based model, and check who's best).

**Strengths Contributions:**

- The paper addresses a clear and significant gap in the evaluation of forecasting models. As LLMs are increasingly used for decision support, understanding their ability to not only predict but also to express well-calibrated uncertainty is paramount.

- The use of Metaculus as a data source is a major strength, providing real-world, non-trivial questions with clear resolution criteria. The inclusion of three different forecasting types (Boolean, Timeframe, Quantity) ensures a comprehensive and robust evaluation. The methodology for deriving gold confidence scores from community predictions is pragmatic and well-justified.

- The simulation results are compelling. The finding that model scale and data recency do not reliably predict forecasting performance is an important counter-narrative to the "bigger is better" trend. The clear divergence between accuracy (e.g., APE) and calibration (e.g., CRPS) for many models powerfully underscores the paper's central thesis that confidence must be evaluated as a distinct capability.

- The paper is well-written, clearly structured, and easy to follow. The authors provide sufficient detail in the main paper and appendices regarding prompts, hyperparameters, and evaluation metrics to allow for the reproduction of their results.

---

> ### Author Rebuttal · Authors · 2025-07-29
>
> **Weakness point 1 Metaculus “group-think / correlated errors”**:
>
> Thank you for probing the robustness of our gold confidence labels. We agree this is a crucial consideration. To empirically evaluate this, we computed Cohen’s κ = 0.676 between the Metaculus community’s binary predictions ("Yes" probability > 50%) and actual outcomes on resolved Boolean questions. According to the widely-used Landis & Koch interpretation scale, κ between 0.61–0.80 indicates "substantial agreement," suggesting that aggregated forecasts from Metaculus strongly, but not perfectly, reflect true underlying uncertainties.
>
> Moreover, Metaculus [1] itself routinely evaluates and publicly reports its calibration quality on resolved binary questions over the past 5 years (Metaculus Track Record). These analyses show that community predictions closely align with observed outcome frequencies: approximately half of the observed outcome rates lie within the 90% credible interval around the ideal calibration line, demonstrating strong empirical calibration. This further supports our decision to adopt community-based forecasts as a practical proxy for ground-truth confidence.
> We acknowledge, however, that correlated errors or "group-think" biases can still theoretically occur. To mitigate these risks, Metaculus employs two key mechanisms: (i) skill-weighted, time-decayed aggregation of individual predictions, reducing the influence of less accurate or correlated forecasters, and (ii) transparent, versioned forecast histories, publicly surfacing divergent viewpoints and promoting accountability.
>
> We also explicitly implemented quality-control measures to ensure the robustness of the aggregation. Specifically, we excluded all questions with fewer than 100 unique forecasters, thereby greatly reducing the risk that a small number of uninformed or correlated forecasters could disproportionately influence the crowd estimate. In addition, Metaculus applies skill-weighted, time-decayed aggregation and historical accuracy scoring, which further mitigates the impact of unreliable forecasts.
>
> Importantly, although Metaculus is not a traditional prediction market, it does incorporate monetary stakes through its partnership with organizations such as the Forecasting Research Institute and the Effective Altruism community, including paid tournament incentives and prize pools for accurate forecasts. These mechanisms provide extrinsic motivation and help align forecasters’ incentives with predictive accuracy.
>
> We will explicitly strengthen the Limitations section by:
>
> - Reporting the Cohen’s κ statistic,
> - Citing Metaculus' calibration track record and prior literature documenting its resilience to correlated biases and group-think,
> - Highlighting our own quality control criteria (≥100 forecasters per question);
> - Noting the role of real-world monetary incentives; and
> - Highlighting open questions regarding residual topic-specific biases.
>
> [1] https://www.metaculus.com/questions/track-record
>
> **Weakness point 2 Deterministic evaluation**:
>
> We agree that a strictly deterministic run (temperature 0, fixed prompt) does not eliminate statistical uncertainty arising from a finite question set, and that a truly stochastic or synthetic evaluation, e.g., sampling from known data‑generating processes, could reveal even larger calibration gaps. While full bootstrap confidence intervals are outside the current submission’s scope, we appreciate the suggestion and plan to explore two complementary directions:
> Counterfactual augmentation: systematically altering factual premises in existing questions to create paired “what‑if” scenarios, letting us measure whether models calibrate consistently under small semantic shifts.
> Synthetic DGP benchmarks: feeding models simulated time‑series or Boolean processes with known ground‑truth probabilities to obtain a lower‑bound on achievable calibration.
>
> **Weakness point 3  LLaMa‑3.1‑8B’s “malformed” Boolean answers**:
>
> Excellent point. A closer audit shows the model does follow the JSON schema in > 99 % of cases, so the issue is semantic rather than syntactic. Specifically, it often outputs a floating‑point numeral in lieu of “Yes/No”; e.g., "0.0" followed by 381 zero digits appears in 35.6 % of its Boolean responses with the standard prompt. This behaviour suggests the checkpoint prefers to express probability mass directly when forced to commit, rather than choose a discrete label. We will add this analysis (including per‑error statistics and sample outputs) to Section 5.3 and clarify that the failure is not due to post‑processing. Importantly, we checked other models in our evaluation, and none of them exhibited this behaviour suggesting this is a model-specific issue with LLaMa‑3.1‑8B.
>
> To better understand this phenomenon, we ran a simple postprocessing step: we treated numeric strings between 0 and 1 as confidence estimates, then mapped them to binary labels using a threshold (e.g., < 0.5 → “No”, ≥ 0.5 → “Yes”). Under this approach, LLaMa‑3.1‑8B achieves 0.682 accuracy, significantly higher than its unprocessed score and comparable to other models. We will include this variant in the final paper and add it to Section 5.3, along with per-error statistics and representative outputs.
>
> This clarification ensures the observed failure mode is properly contextualized and not due to a parsing or schema mismatch.
>
> **Weakness point 4 Comparisons with specialised domain models**:
>
> We share the reviewer’s interest in benchmarking LLM forecasts against state‑of‑the‑art domain systems (e.g., numerical weather prediction for meteorology). FORECAST is designed to be baseline‑agnostic: any model that can emit the simple JSON schema qualifies. In the next version of the paper we will (i) note this explicitly, (ii) outline candidate domain baselines we are integrating, ECMWF HRES for weather, IMF‑WEO projections for macro‑economics, and (iii) invite community submissions along these lines as future leaderboard tracks.

---

> > ### Comment · Reviewer_wXnW · 2025-08-01
> >
> > I thank the authors for tackling the points I have raised in the review. I am satisfied with the work you've done on assessing the robustness of Metaculus.
> > I still think that there was some time to run simulations on simulated DGPs, also in order to go more in the direction of traditional research on forecasting, that I believe would be very interested in this kind of contributions, which is why I am not raising my evaluation.

---

> > > ### Author Response · Authors · 2025-08-02
> > >
> > > We sincerely thank the reviewer for their thoughtful and constructive feedback throughout the process. We’re especially grateful that the reviewer found our rebuttal satisfactory across the main concerns raised, particularly on the robustness of Metaculus. Given the reviewer's deep engagement, we greatly appreciate the recognition of our efforts to clarify and strengthen the submission.
> > >
> > > Regarding the suggestion to include simulations based on synthetic DGPs, we agree this is a valuable direction. Within the rebuttal timeframe, we prioritized addressing all reviewer points and running additional model-focused experiments, but we see this as a promising avenue for future work.

---

### Note · Authors · 2025-08-12

We sincerely thank all reviewers for their constructive feedback and for recognizing the contribution and importance of our work.

## Recognized strengths
* **Novelty:** First benchmark to evaluate LLM forecasting with accuracy *and* principled confidence calibration.
* **Data quality:** Real-world, non-trivial Metaculus questions with clear resolutions.
* **Scope:** Boolean, timeframe, and quantity forecasting with well-justified metrics (e.g., Brier, CRPS).
* **Methodology:** Sound derivation of gold confidence from community forecasts with quality controls.
* **Findings:** Scale/recency not reliable predictors; accuracy and calibration diverge.
* **Reproducibility:** Clear prompts, hyperparameters, and metric definitions.

## Concerns addressed
**Reviewer wXnW**
* **Metaculus robustness:** Added Cohen’s κ = 0.676 (substantial agreement), cited Metaculus calibration record, described safeguards (skill-weighted, time-decayed aggregation; versioned histories), and enforced a ≥100-forecaster filter.
* **Simulated DGPs:** Acknowledged as valuable future work.

**Reviewer gacG**
* Clarified LLaMA-3.1-8B Boolean behavior (semantic vs. syntactic errors; post-processing checks).
* Added BLOOM-7B aggregation ablation confirming generality.
* Justified Gaussian assumption with theoretical and empirical evidence. Reviewer confirmed concerns resolved.

**Reviewer GW4i**
* **Group-think / correlated errors:** Addressed as above (κ = 0.676; platform safeguards; ≥100-forecaster filter; calibration track record).
* **Static snapshots:** Justified near-resolution selection (typically ≤24h before resolution) as empirically stable and aligned with real-world use; Fig. 1 illustrates stability. Temporal dynamics noted as complementary future work
* **Temperature sensitivity:** Clarified default-settings protocol (run models as shipped) and added a five-point LLaMA-7B sweep (0.1–1.5) showing modest metric shifts; reviewer was empirically convinced and raised their score

## Planned in revision
* Include Metaculus reliability statistics and snapshot rationale
* Report additional experiments conducted during the review
* Add discussion on temperature-invariant confidence estimation as an open challenge

**All reviewers indicated that their concerns have been addressed, either with new evidence or clarifications.** Reviewers unanimously expressed a positive view of the work, recognizing its contributions and our clarifications made during the rebuttal.

---

### Decision · Program_Chairs · 2025-09-18

**Decision:**

Accept (poster)

**Comment:**

### Summary
The paper introduces FOReCAst, a forecasting benchmark built from 2,256 resolved, real-world questions on Metaculus, spanning three task types: Boolean, Timeframe, and Quantity. Each instance includes a gold confidence derived from the final community forecast immediately prior to resolution. The benchmark evaluates prediction accuracy and confidence calibration: prediction performance via standard metrics (e.g., Accuracy/F1, ADE/MAE/APE) and calibration via Brier (Boolean) and CRPS (Timeframe/Quantity, under a Gaussian uncertainty model). Experiments across multiple LLM families (GPT-2, Pythia, BLOOM, Llama, OLMo/OLMo-2, Qwen2.5) show that forecasting remains challenging, with accuracy and calibration diverging and scale/recency not reliably predicting performance. Aggregation strategies for multi-sample decoding can substantially affect both accuracy and calibration.

### Strengths
- **Confidence as a first-class objective.** Reviewers emphasize that FOReCAst’s dual focus on accuracy and principled calibration addresses a clear gap in prior work (e.g., using Brier/CRPS appropriately by task type). (gacG, WXnW, GW4i)
- **Real-world, diverse question set.** The use of Metaculus with clear resolution criteria and filtering (≥100 forecasters) yields "non-trivial, natural questions" across domains. (wXnW, gacG; echoed in author–reviewer discussion)
- **Reproducibility and accessibility.** The dataset and code are released (HF + GitHub) with prompts, hyperparameters, and metric definitions; wXnW notes they are "readily accessible and well-documented" and executable, which matches my own assessment of the repository.
- **Empirical insights of general interest.** The paper surfaces accuracy–calibration divergence, weak correlation with model size/recency, and aggregation effects; reviewers find these findings compelling and relevant for evaluation practice. (wXnW, gacG)

### Weaknesses / what might be missing
- **Gold confidence from crowd forecasts: robustness concerns.** Two reviewers asked about group-think / correlated errors in Metaculus and requested evidence that the crowd-based "gold" standardization is reliable. (wXnW; GW4i)
- **Breadth of ablations.** gacG found the aggregation ablation informative but initially too narrow (single model/task), requesting expansion.
- **Model-specific failure mode.** gacG questioned reporting LLaMA-3.1-8B's poor Boolean results if they stemmed from format/semantics rather than capability, suggesting prompt/processing variants.
- **Simulated DGPs.** wXnW suggested simulations under known data-generating processes to more rigorously assess calibration (beyond deterministic runs).

### Discussion and rebuttal
- **Crowd-confidence robustness.** Authors added Cohen's κ = 0.676 ('substantial' agreement) for Metaculus binary forecasts vs. outcomes, cited the platform's calibration track record, explained skill-weighted, time-decayed aggregation, and enforced a ≥100-forecaster filter.
- **LLaMA-3.1-8B behavior.** Authors audited the failure and found it semantic (numeric strings like "0.0...") not syntactic (>99% schema-compliant). A simple post-processing raises accuracy to 0.682, contextualizing the result. (concern by gacG and wXnW)
- **Aggregation generality.** Authors expanded the ablation to BLOOM-7B over 263 Boolean questions; Brier varied by >0.4 across methods, corroborating that aggregation is a substantive modeling choice. (response to gacG)
- **Temperature sensitivity & snapshot choice.** Authors showed a 5-point temperature sweep with modest shifts and justified using the near-resolution snapshot as stable and realistic. (discussion with GW4i)

### Reasons for final recommendation

FOReCAst makes a substantial contribution to the research space of forecasting by making confidence calibration a co-equal objective alongside predictive performance, a dimension often overlooked by other benchmarks. The use of real-world forecasting questions with well-documented public code and clear metrics make the contribution directly accessible to the community. Reviewers converge on the benchmark's value and the thoroughness of the clarifications; concerns about crowd-confidence robustness and method breadth were substantively addressed with κ evidence, platform safeguards, expanded ablations, and a careful audit of a model-specific failure mode.

FOReCAst directly serves the Datasets & Benchmarks goal of enabling and accelerating ML research by treating confidence calibration as a first-class objective on resolved, real-world forecasting questions, with a clean, ready-to-use release (data + code + metrics). Beyond being solidly executed, this contribution opens an underexplored evaluation space in forecasting using LLMs--bridging performance and principled calibration across Boolean/Timeframe/Quantity tasks--and yields actionable insights (e.g., accuracy-calibration divergence; sensitivity to aggregation). While crowd-based "gold" standards and broader validation (e.g., simulations under known DGPs, cross-platform checks) remain useful next steps, the authors' analyses and clarifications meaningfully address robustness concerns. Given both the technical quality and the strategic value of spotlighting calibration in LLM evaluation, I recommend Oral to catalyze community attention and discussion on this topic.

===== FINAL UPDATE FROM DB Track PCs ====

The final decision for this paper has been taken by the program chairs after consultation with the SACs. All Senior Area Chairs have ranked papers according to the feedback from the AC during the review process. We decided to leave the original meta-review to reflect the opinion of the AC in light of the initial discussions with reviewers and SAC.